# Oriented intergrowth of the catalyst layer in membrane electrode assembly for alkaline water electrolysis

Lei Wan [1], Maobin Pang[1], Junfa Le[1], Ziang Xu[1], Hangyu Zhou [1], Qin Xu[1] & Baoguo Wang [1] ✉

The application of membrane electrode assemblies is considered a promising approach for increasing the energy efficiency of conventional alkaline water electrolysis. However, previous investigations have mostly focused on improving membrane conductivity and electrocatalyst activity. This study reports an all-in-one membrane electrode assembly obtained by de novo design. The introduction of a porous membrane readily enables the oriented intergrowth of ordered catalyst layers using solvothermal methods, leading to the formation of an all-in-one MEA for alkaline water electrolysis. This all-in-one MEA features ordered catalyst layers with large surface areas, a low-tortuosity pore structure, integrated catalyst layer/membrane interfaces, and a well-ordered OH⁻ transfer channel. Owing to this design, a high current density of 1000 mA cm⁻² is obtained at 1.57 V in 30 wt% KOH, resulting in a 94% energy efficiency. This work highlights the prospects of all-in-one membrane electrode assemblies in designing next-generation high-performance alkaline water electrolysis.

The conversion and storage of sustainable clean energy is an effective approach for solving the energy crisis and environmental issues faced by modern society. Hydrogen production by water electrolysis has attracted enormous interest from both academia and industry[1,2]. As a commercially mature hydrogen production technology, the conventional alkaline electrolyser is characterised by two electrodes immersed in a concentrated alkaline solution (20–30 wt.%)[3]. A porous membrane permeable to hydroxide ions limits gas permeability. However, conventional alkaline water electrolysers have certain critical disadvantages. Gas bubble formation and the thick porous membranes employed cause ohmic loss, which significantly reduces the energy efficiency of conventional alkaline water electrolysers and severely limits high current density performance[4].

To further improve the cost-competitiveness of conventional alkaline water electrolysers, increasing the current density without decreasing the efficiency is an attractive approach[5,6]. Inspired by proton exchange membrane water electrolysis (PEMWE) and fuel cell technology, a zero-gap configuration employing a membrane electrode assembly (MEA) component has been proposed for alkaline electrolysers[7]. Owing to the fact that the electrodes are directly adjacent to the porous membrane, ohmic losses across the electrolyte are significantly decreased, providing an approach for achieving operation at higher current densities. Traditional MEA preparation methods include catalyst-coated substrate (CCS) and catalyst-coated membrane (CCM) techniques. Typically, a catalytic ink containing catalyst powder, ionomer and solvent, is sprayed on a porous gas/liquid diffusion layer or membrane[8]. The catalytic layer structure obtained with the catalytic ink has some critical disadvantages (Fig. 1a). First, the dense catalytic structure impedes gas/liquid mass transfer and decreases effective electrocatalyst utilisation. Second, the ionomer reduces the electrical conductivity and occupies some catalytic active sites. Finally, some catalyst falls off the electrode at high current densities, leading to low electrolysis stability[9]. To solve these drawbacks, some in-situ catalytic layer (CL) deposition methods, such as electrodeposition and hydrothermal and chemical vapour deposition, have been proposed for MEA-CCS with a 3D and binder-free catalytic layer structure[10,11]. However,

[1]Department of Chemical Engineering, Tsinghua University, Beijing, China. ✉e-mail: bgwang@tsinghua.edu.cn

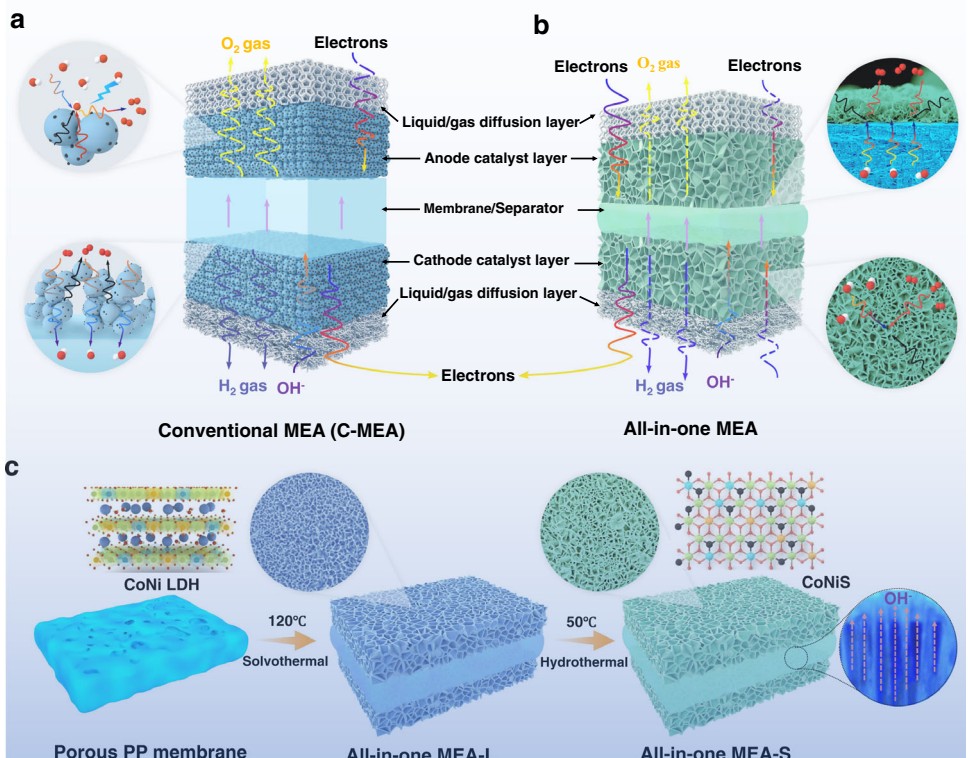

**Fig. 1 | Schematic illustration of the all-in-one MEA concept and preparation process. a** Conventional MEA (C-MEA) and **b** all-in-one MEA based on 3D-ordered transport highways. **c** Synthetic procedure for the all-in-one MEA-S.

these in-situ CL deposition techniques cannot be used to form MEA-CCM on polymer membranes owing to the nonconducting, swelling and thermolabile properties of polymer membranes[12–14]. Furthermore, MEA-CCM ensures good contact between the CL and membranes, decreasing CL/membrane interfacial resistance and improving interfacial stability[15]. In contrast, the hot-pressing process can be utilised to assemble CCS on membranes to achieve good CL/membrane interfacial contact in MEA-CCS (Supplementary Figs. 1 and 2). Therefore, the development of MEAs with both highly ordered CLs and integrated CL/membrane interfaces for advanced alkaline water electrolysis is both highly desirable and challenging.

In this study, an all-in-one MEA with 3D-ordered transport highways and integrated CL/membrane interfaces is designed, with 3D-ordered CLs integrated on a porous polypropylene (PP) membrane (Fig. 1b). The as-prepared MEA has advantages, including: (i) a 3D-ordered CL structure that enhances liquid/gas mass transfer; (ii) CLs with enlarged active sites and increased electronic conductivity owing to the fact that no polymer binder is used; and (iii) decreased interfacial resistance and accelerated ionic transport at interfaces owing to the integrated CL/membrane interface structure. Therefore, advanced alkaline water electrolysers employing this all-in-one MEA show a high current density of 1000 mA cm$^{-2}$ at 1.57 V in 30 wt% KOH. Importantly, the all-in-one MEA operated stably at a high current density of 1000 mA cm$^{-2}$ at 60 °C for more than 1000 h in 30 wt% KOH.

## Results

In the last decade, a considerable amount of research has focused on preparing uniform and dense CLs on membrane surfaces[16,17]. However, the direct construction of CLs with ordered and high-porosity architectures on ion-exchange membrane surfaces is highly challenging. Therefore, we considered whether 3D-ordered CLs could be directly fabricated from porous membranes. To achieve this, alkali-resistant PP porous membranes were used owing to their high porosity and interconnected pore structures (Supplementary Fig. 3). CoNi layered

double hydroxide (LDH) nanosheet arrays were first intergrown on both sides of a porous PP membrane (all-in-one MEA-L) through a facile solvothermal method (Fig. 1c). The CoNi LDH nanosheet arrays were then transformed into CoNiS via mild sulfurization treatment (all-in-one MEA-S). Figure 2a shows a photograph of the all-in-one MEA-S over a large area (20 cm × 100 cm). The colour of the all-in-one MEAs changed from pink to black after sulfurization treatment (Supplementary Fig. 4). Scanning electron microscopy (SEM) analysis was conducted to observe the CL morphologies of the all-in-one MEAs. As shown in Fig. 2b and Supplementary Fig. 5, CoNi LDH nanosheet arrays vertically grew on the entire PP membrane surface. As shown in Fig. 2c, the main structure of the nanosheet arrays after sulfurization was almost unchanged. Such a structure can obviously increase the effective surface area of the CLs and promote electron and ion transmissions, improving the electrochemical performance of the CLs[18]. The uniform nanosheet structures of CoNi LDH and CoNiS were confirmed by transmission electron microscopy (TEM; Fig. 2d and Supplementary Figs. 6 and 7). The measured lattice spacings between the two fringes in Fig. 2e were 0.193 and 0.291 nm, which were attributed to diffraction by the (422) lattice plane of $Ni_3S_4$ and the (311) lattice plane of $Co_9S_8$, respectively[19]. Furthermore, energy-dispersive X-ray (EDX) elemental mapping images showed that Co, Ni and S were evenly distributed in the CoNiS nanosheets (Fig. 2f). The thickness of the CoNiS nanosheets was determined through atomic force microscopy (AFM) analysis (Supplementary Fig. 8). The average thickness of the CoNiS nanosheets was ~4.0 nm. The X-ray powder diffraction (XRD) patterns and X-ray photoelectron spectrometry (XPS) results of the CoNiS catalysts are shown in Supplementary Figs. 9 and 10.

Figure 2g shows the cross-sectional morphology of the all-in-one MEA-S based on employing the CoNiS nanosheet arrays, which were vertically aligned with the separator, as CLs. The CL thickness was estimated to be 3 μm. The average pore diameters of the CLs in the all-in-one MEA-L (1.22 ± 0.28 μm) and all-in-one MEA-S (1.52 ± 0.31 μm) were significantly higher than that of a conventional

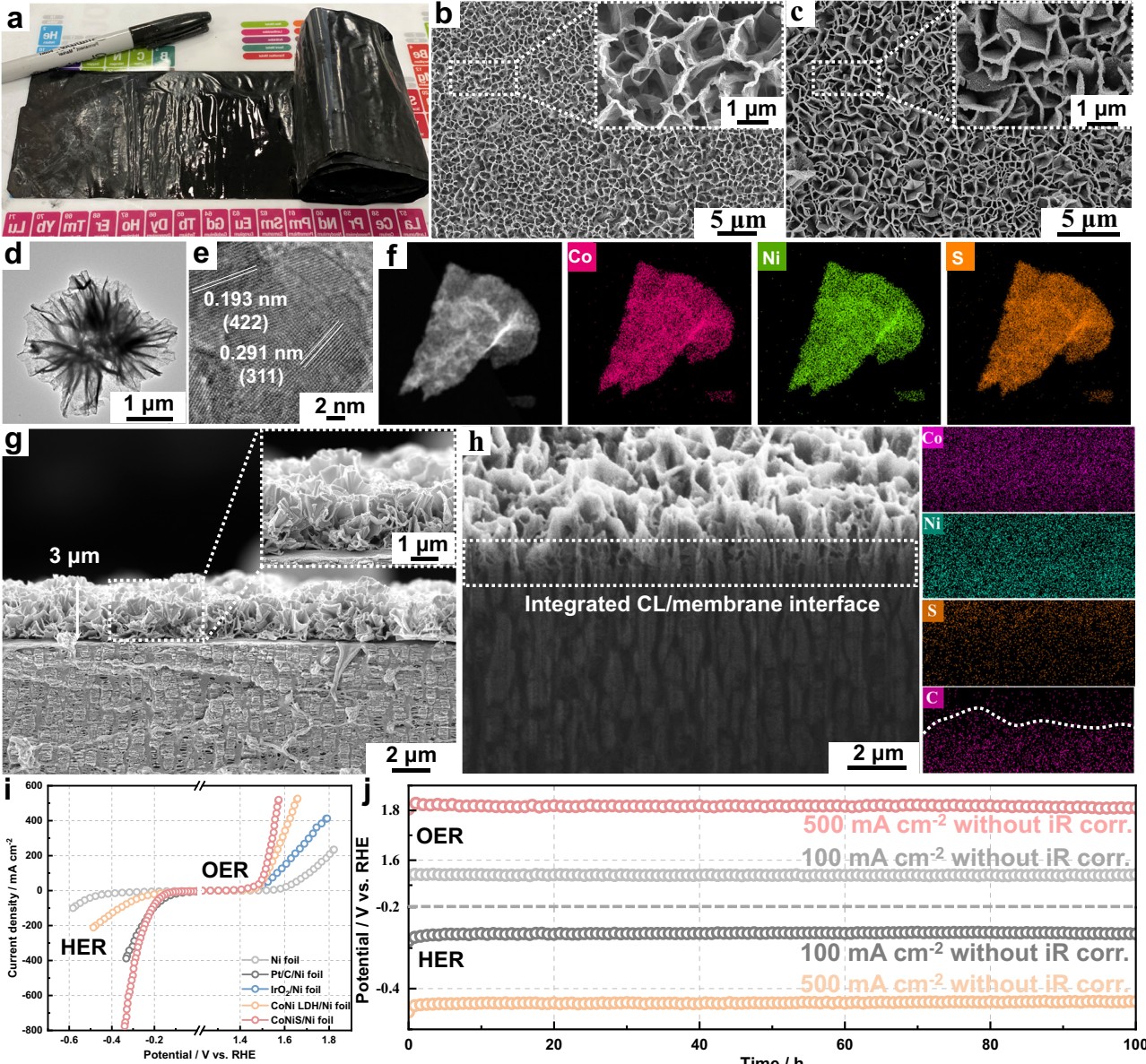

**Fig. 2 | Morphological and catalytic activity characterisation of the all-in-one MEAs. a** Optical image of the all-in-one MEA-S. SEM images of the **b** all-in-one MEA-L surface and **c** all-in-one MEA-S surface. **d** TEM and **e** HRTEM images of the CoNiS nanosheets. **f** Elemental mapping images of the CoNiS nanosheets. **g** Cross-sectional SEM image of the all-in-one MEA-S. **h** Cross-sectional SEM image of the CL in the all-in-one MEA-S prepared by a focused ion beam and corresponding EDX mapping of the CL cross-section. **i** Linear sweep voltammetry (LSV) curves of the electrodes evaluated by a standard three-electrode system for the OER and HER. **j** Chronopotentiometric curves of the CoNiS electrode at constant current densities of 100 and 500 mA cm$^{-2}$ for 100 h.

MEA (C-MEAs)-CCM with dense CLs (Supplementary Figs. 11 and 12). The porosities of the CLs in the all-in-one MEA-L and all-in-one MEA-S were 90% and 92%, respectively, compared with that (only 5%) of conventional CLs prepared by the CCM method. Porous, ultrathin and binder-free CLs are beneficial for increasing the electrochemical active area and enhancing gas/liquid mass transport[20,21]. Supplementary Figure 13 shows photographs and the CL morphology of the all-in-one MEA-S after stretching, indicating that the CLs could be flexed without undergoing structural damage. To further investigate the CL/membrane interface structure, the cross-sectional morphology of the all-in-one MEA-S was observed after treatment by a focused ion beam (Fig. 2h). Interestingly, an integrated CL/membrane interface structure was observed, which accelerated OH⁻ mass transfer at the interface[22,23]. As expected, the CL also grew into the porous channels of the PP membrane, strongly connecting the CL to the PP membrane[24]. In addition, strong

adhesion between the membrane and CoNiS nanosheet arrays was observed, as evidenced by the CoNiS nanosheets not detaching from the material after sonication for 10 min (Supplementary Fig. 14). In sharp contrast, under strong ultrasonic vibration, a large number of Pt/C and IrO$_2$ nanoparticles detached from the C-MEA-CCM surface comprising a commercial anion exchange membrane (AEM), resulting in the destruction of the CL structure (Supplementary Fig. 15). The interfacial bond strengths of the C-MEA-CCM and the all-in-one MEA-S were also quantitatively compared (Supplementary Fig. 16). The average interfacial bonding strength was 7.9 mN mm$^{-1}$ for C-MEAs-CCM, but was significantly higher for the all-in-one MEA-L (20.0 mN mm$^{-1}$) and all-in-one MEA-S (23.0 mN mm$^{-1}$). This result indicated that the all-in-one MEA structure can withstand large interfacial stress because nanosheet networks grow into the porous channels of the PP membrane, strongly connecting CLs to the PP composite membrane.

To understand how 3D-ordered nanosheet arrays form inside and on the surface of the porous PP membrane, various experimental parameters were investigated to observe the 3D-ordered catalytic membrane architecture. First, to elucidate the formation process of the 3D-ordered CLs, the morphologies of the all-in-one MEA-S after solvothermal treatment for different time periods were observed by SEM. The cross-section SEM images suggested that CoNi LDH nanosheets grew inside the porous PP membrane as the solvothermal treatment time increased (Supplementary Fig. 17). When the solvothermal treatment time exceeded 4 h, almost all visible pores were filled with CoNi LDH nanosheets. The CL surface morphologies were further observed by adjusting the solvothermal treatment time (Supplementary Fig. 18). A small number of nanoparticles was clearly observed on the surface of the porous PP separator when the solvothermal treatment time was less than 4 h. With a solvothermal treatment time of 4 h, isolated CoNi LDH nanosheets grew from the inside of the porous PP membrane. After 5 and 6 h, the CoNiS nanosheets further grew to form networks, uniformly covering on the entire surface of the porous PP membrane. As the solvothermal treatment time further increased, the thickness of the CoNi LDH nanosheets gradually increased, resulting in decreased CL porosity. Therefore, the growth mechanism of the 3D-ordered catalytic membrane architecture was similar to the inside-out germination of natural trees from ploughlands (Supplementary Fig. 19). To further understand the growth mechanism of the 3D-ordered catalytic membrane architecture, different membranes were used for solvothermal preparation (Supplementary Fig. 20). The 3D-ordered CLs did not form on the surface of an AEM (FAA-3-20) with a dense membrane structure. Furthermore, the AEM exhibited severe swelling behaviour caused by the ethanol solvent. Moreover, when water was used as the solvent instead, the 3D-ordered CLs still did not form on the AEM or PP membrane surface (Supplementary Fig. 21). To further eliminate the swelling factor, a poly(2,2'-(m-phenylene)−5,5'-bibenzimidazole) (m-PBI) membrane without cations was used. However, 3D-ordered CLs also failed to form on the m-PBI membrane surface. However, the 3D-ordered CLs could grow on two sides of a porous polyethylene (PE) separator. These results indicate that the porous membrane structure plays an indispensable role in the formation of 3D-ordered CLs.

CLs with high activity and durability towards the oxygen evolution reaction (OER) and hydrogen evolution reaction (HER) are necessary to achieve high-performance alkaline water electrolysis. The catalytic performance of CLs loaded on a polymer separator can not be evaluated by a standard three-electrode system. Therefore, the PP separator was replaced with Ni foil to prepare CoNiS/Ni foil. For comparison, catalysts including bare Ni foil, CoNi LDH, Pt/C and IrO$_2$ were loaded on Ni foil (Supplementary Fig. 22). The linear sweep voltammetry (LSV) curves (Fig. 2i) showed increasing OER activity in the order of bare Ni foil <IrO$_2$ < CoNi LDH < CoNiS. Notably, the CoNiS/Ni foil featured low overpotentials of 195 and 280 mV to attain current densities of 10 and 100 mA cm$^{-2}$, respectively (Supplementary Fig. 23). Next, the HER activities of these samples were assessed. The CoNi LDH exhibited moderate HER activity. In contrast, sulfurization treatment significantly improved the HER activity of CoNi LDH, especially at high current densities. Specifically, CoNiS delivered low overpotentials of 121 and 210 mV to generate cathodic current densities of 10 and 100 mA cm$^{-2}$, respectively (Supplementary Fig. 24). In addition, when Ni foil was replaced with porous Ni foam, the obtained electrochemical performance of the electrodes was almost the same (Supplementary Fig. 25). Electrochemical impedance spectroscopy (EIS) and electrochemically active surface area (ECSA) analysis were used to determine the origin of the excellent catalytic activity of CoNiS (Supplementary Figs. 26−29). The results demonstrated that CoNiS had much higher intrinsic activity towards the OER and HER compared with CoNi LDH. Durability plays a vital role in evaluating the suitability of a bifunctional catalyst. Figure 2j shows the results of the long-term stability tests

performed on the CoNiS nanosheet arrays. The CoNiS nanosheet arrays showed negligible changes in potential at 100 and 500 mA cm$^{-2}$ after 100 h of operation (Supplementary Fig. 30). Changes in the CL morphology were further investigated after the stability tests (Supplementary Figs. 31 and 32). The morphology and element composition of the CoNiS nanosheet arrays were observed to be almost unchanged. Furthermore, the overpotential values of the CoNiS nanosheet arrays were comparable or smaller than those of recently reported bifunctional catalysts towards the HER and OER (Supplementary Fig. 33). The high activity and durability of CoNiS indicate its promising application potential as a bifunctional electrocatalyst for advanced alkaline water electrolysis.

In general, pristine porous PP membranes cannot be used for water electrolysis owing to their high gas permeability and low ionic conductivity. The pristine PP membrane exhibited a thickness of 20 μm and typical elliptical pore structures, as shown in Fig. 3a. After in-situ pore-filling treatment, the all-in-one MEA-S showed a thickness of 26 μm, and micropores densely filled with CoNiS nanosheets (Fig. 3b). Cross-sectional EDS mapping showed that the CoNiS nanosheets were uniformly distributed in the through-plane direction of the composite membrane. X-ray microcomputed tomography (μ-CT) was further used to investigate the interior structure and homogeneity of the composite membrane on a larger scale. As shown in Figs. 3c, d, the structure of the composite membrane showed fewer pores and cracks in all cross-sectional diagrams compared with the pristine PP separator, demonstrating that the CoNiS nanosheet networks uniformly and densely formed in the PP substrate on a relatively macroscopic scale. Mercury intrusion porosimetry (MIP) and capillary flow porometry (CFP) were also used to analyse the pore structures of the all-in-one MEA-S (Supplementary Figs. 34 and 35). The all-in-one MEA-S exhibited an average pore diameter of ~30 nm, which was significantly lower than those of PP (~0.30 μm) and commercial Zirfon Thin (~0.45 μm) membranes (Supplementary Table 1). The liquid contact angle of the membranes is shown in Fig. 3e. Electrolyte penetration into the pristine PP and commercial Zirfon Thin membranes was difficult, while for the all-in-one MEA-S, entire liquid droplets spread into the composite membrane almost instantaneously. This phenomenon was attributed to the inherent hydrophilicity of the metal sulfide, which enhanced the wettability of the composite membranes[4,25]. To avoid gas crossover occurring by mixing with a gas-saturated electrolyte, good gas barrier properties for membranes are indispensable[26,27]. Therefore, the bubble point pressure (BPP) was used to assess the membrane gas permeability. The BPP of the composite membrane in the all-in-one MEA-S reached 11.1 bar, which was significantly higher than that of the pristine PP (1.0 bar) and commercial Zirfon (2.5 bar) membranes. This high BPP was attributed to the extensive incorporation of nanosheet networks into the robust polymer matrix[4]. Figure 3f shows the area resistance of commercial membranes and the all-in-one MEA-S as a function of KOH solution concentration. The all-in-one MEA-S exhibited a lower area resistance in KOH solutions of different concentrations than the Zirfon Thin, m-PBI and Sustainion X37-50 membranes. The area resistance of the commercial Zirfon membrane and m-PBI membrane sharply increased from 0.30 and 0.27 Ω cm$^2$ in 30 wt% KOH solution to 1.50 and 1.37 Ω cm$^2$ in 0.5 wt% KOH solution. In contrast, the all-in-one MEA-S showed a weakly increasing area resistance trend with decreasing electrolyte concentration. The all-in-one MEA-S showed the lowest area resistances of 0.30 and 0.05 Ω cm$^2$ in 0.5 and 30 wt% KOH solutions, respectively, which were even lower than those of the Sustainion X37-50 membrane (0.32 and 0.28 Ω cm$^2$). To gain further insight into OH$^-$ transport behaviour in the composite membrane, the ion transference number throughout the membrane was analysed and calculated, as shown in Fig. 3g. Compared with the ion transference numbers of the commercial Zirfon Thin, m-PBI and Sustainion X37-50 membranes, that of the composite membrane in the all-in-one MEA-S

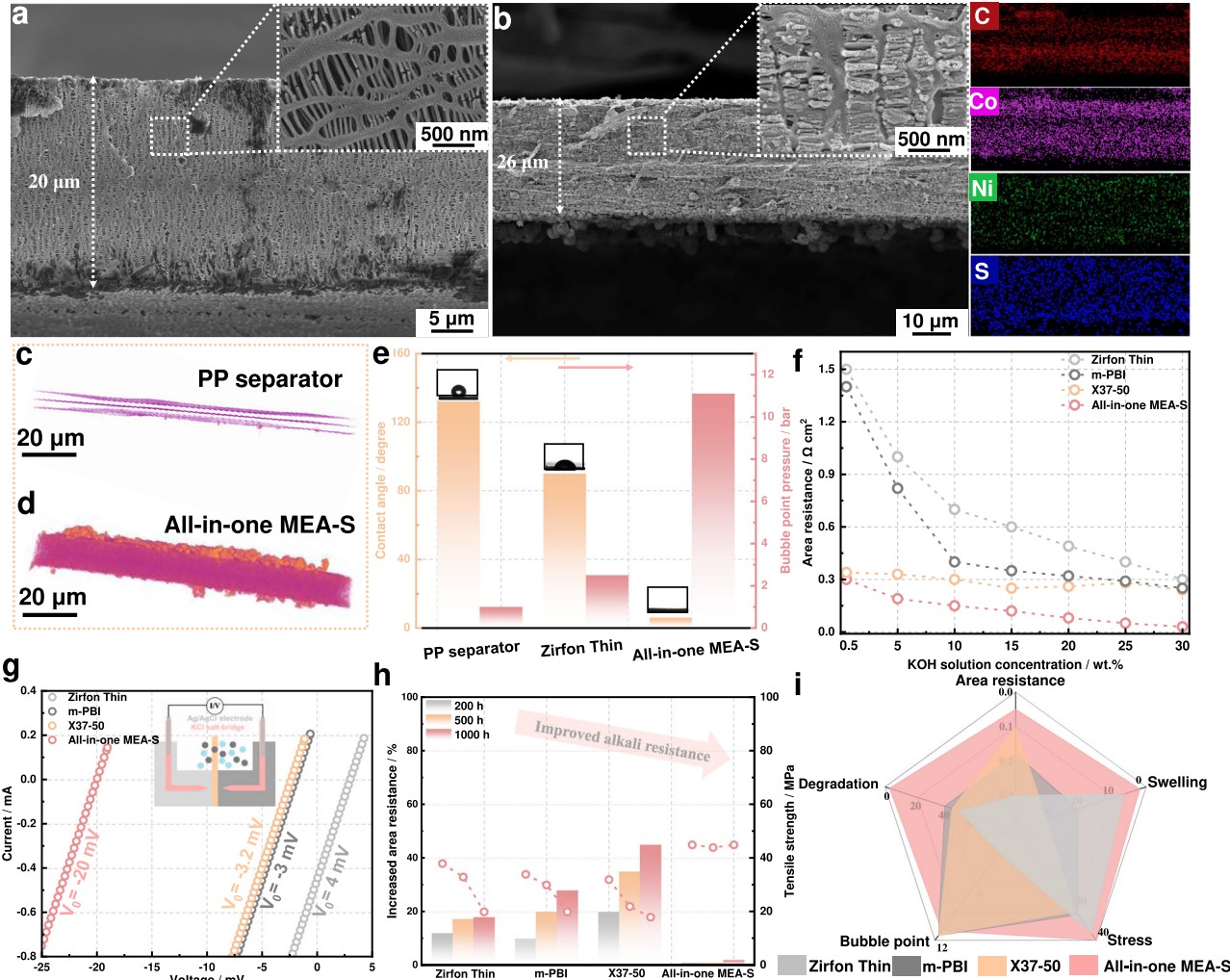

**Fig. 3 | Basic properties inferred from membrane characterisation.** Cross-sectional SEM images of the **a** pristine PP porous separator and **b** all-in-one MEA-S, and corresponding elemental mapping. Inset: high-resolution SEM images. The 3D-reconstruction image projection of the **c** pristine PP porous separator and **d** all-in-one MEA-S. **e** Contact angle and bubble point pressure of the pristine PP porous separator, Zirfon Thin and all-in-one MEA-S. **f** Area resistance of the commercial Zirfon Thin, *m*-PBI and Sustainion X37-50 membranes, and all-in-one MEA-S. **g** Current-voltage profiles of the Zirfon Thin, m-PBI, Sustainion X37-50 membranes and all-in-one MEA-S measured using a KOH concentration gradient of 1–3 mol L$^{-1}$. **h** Area resistance and mechanical strength of different membranes at different treatment times of 200, 500 and 1000 h in 30 wt% KOH at 80 °C. **i** Comprehensive comparison of the properties of different membranes.

indicated high hydroxide ion conductivity (see "Methods section" for details).

The different responses of area resistance with changing electrolyte concentration were attributed to different OH$^-$ transport mechanisms. For example, OH$^-$ conduction in the Sustainion X37-50 membrane with ion-exchange functionality can occur via a combination of Grotthuss and vehicular mechanisms, causing the OH$^-$ conductivity to be largely related to the nature of tethered cationic groups, ion-exchange capacity and the tortuosity of the OH$^-$ conduction pathway[28,29]. Therefore, the external alkaline electrolyte has a weak impact on the area resistance of AEMs. The *m*-PBI membrane utilises the uptake of KOH electrolyte to achieve hydroxide conductivity, and is not necessarily an intrinsic hydroxide conductor[30]. Therefore, the area resistance of *m*-PBI membranes is largely determined by the supporting electrolyte. For porous membranes, including the Zirfon Thin and as-prepared PP composite membranes, the electrolyte was transported through the diaphragm pores. The swelling and electrolyte behaviour of the commercial membranes and all-in-one MEA-S were also evaluated (Supplementary Fig. 36). Notably, the as-prepared PP composite membrane showed superior antiswelling behaviour (0% swelling in 30 wt% KOH). The alkaline stability of the

membranes is also important for long-term electrolysis operation owing to OH$^-$ attack[31]. The ex-situ durability under alkaline conditions was used to screen membranes for long-term operation under alkaline water electrolysis. The alkaline stability of the composite membrane was studied in 30 wt% KOH at 80 °C (Fig. 3h), with the all-in-one MEA-S showing a negligible change in area resistance after alkaline treatment for 1000 h. As expected, the excellent chemical stability of the composite membrane in the all-in-one MEA-S resulted from the ether-free polymer backbone and chemically stable metal sulfide[32,33]. However, the commercial Zirfon Thin, *m*-PBI and Sustainion X37-50 membranes exhibited 18%, 30% and 45% increases in area resistance after durability testing for 1000 h. Furthermore, the change in the mechanical properties of the membranes after alkaline stability testing was investigated. The commercial FAA-3-20 and m-PBI membranes lost 80% (from 25 to 5 MPa) and 47% (from 36 to 20 MPa) of their tensile strength after durability testing for 1000 h (Supplementary Fig. 37). In contrast, the composite membrane exhibited a negligible change in mechanical properties. Finally, as shown in Fig. 3i, the all-in-one MEA-S exhibited overwhelming advantages for alkaline water electrolysis over representative commercial membranes in terms of area resistance, swelling, stress strength, alkaline stability and gas barrier properties.

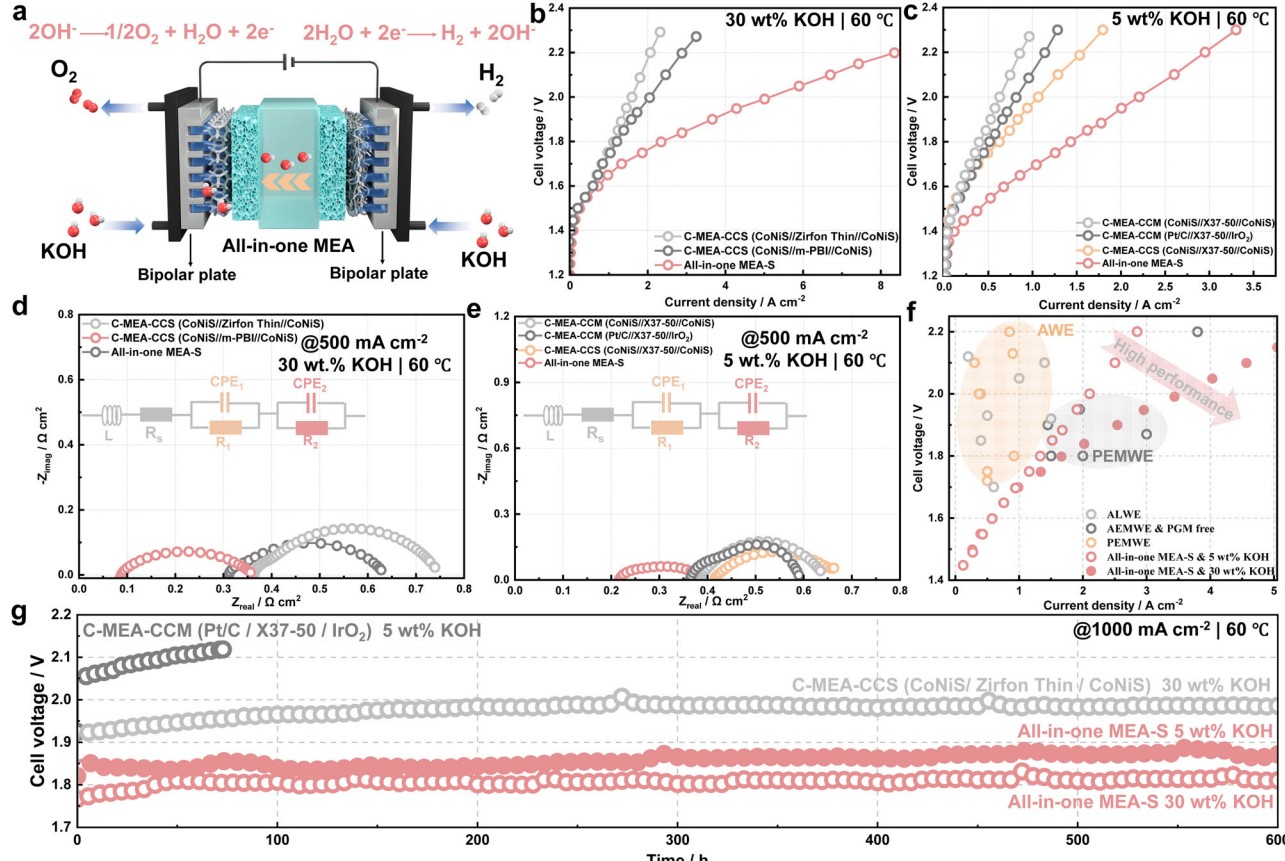

**Fig. 4 | Electrolysis performance of the all-in-one MEAs. a** Schematic diagram of advanced alkaline water electrolysis using the all-in-one MEA. Carbon paper and Ni foam were used as the cathode and anode liquid/gas diffusion layers, respectively. **b** Polarisation curves of the C-MEA-CCS (CoNiS//Zirfon Thin//CoNiS), C-MEA-CCS (CoNiS//m-PBI//CoNiS) and all-in-one MEA-S in 30 wt% KOH at 60 °C. The C-MEA-CCS (CoNiS//Zirfon Thin//CoNiS) and C-MEA-CCS (CoNiS//m-PBI//CoNiS) were prepared using self-supported CoNiS electrodes. **c** Polarisation curves of the C-MEA-CCM (CoNiS//X37-50//CoNiS), C-MEA-CCM (Pt/C//X37-50//IrO₂), C-MEA-CCS (CoNiS//X37-50//CoNiS) and all-in-one MEA-S in 5 wt% KOH at 60 °C. The C-MEA-CCS (CoNiS//X37-50//CoNiS) was prepared using self-supported CoNiS electrodes. **d** Nyquist impedance plots of different MEAs in 30 wt% KOH at 60 °C. **e** Nyquist impedance plots of different MEAs in 5 wt% KOH at 60 °C. **f** Comparison of advanced alkaline water electrolysis using the all-in-one MEA-S with reported ALWE, PGM-free AEMWE and PEMWE. **g** Durability cell voltage-time plots for the C-MEA-CCM and all-in-one MEA-S at a constant current density of 1000 mA cm⁻² at 60 °C.

Advanced alkaline water electrolysis was performed by coupling the all-in-one MEAs with liquid/gas diffusion layers (Fig. 4a), and the performance was compared with that of C-MEAs prepared by the CCS or CCM method. Figure 4b compares the alkaline water electrolysis performance of the all-in-one MEA-S and the alkaline liquid water electrolysis (ALWE) performance of the C-MEA-CCS in 30 wt% KOH. The current density of the all-in-one MEA-S was 2.5 A cm⁻² at 1.8 V, representing a 2.5-fold increase compared with that of C-MEA-CCS using the Zirfon Thin and $m$-PBI membranes. The corresponding EIS spectrum (Fig. 4d) were used to investigate the contributions of ohmic and charge-transfer resistances. A perfect semicircle was obtained for the all-in-one MEAs, whereas the presence of a semicircle at low frequency originating from mass-transfer resistance was observed for the C-MEA-CCM[34]. The impedance at high frequency, intercepting the $x$-axis, represented the cell ohmic resistance ($R_{ohm}$)[35]. The estimated initial area-specific resistances of the all-in-one MEA-S and C-MEA-CCS comprising the Zirfon Thin and $m$-PBI membranes were equal to 0.09, 0.30 and 0.35 Ω cm² in 30 wt% KOH, respectively, which is in good agreement with the ex-situ area resistance data. The cell ohmic resistance results also suggested that the electrolyser was not a short circuit. The all-in-one MEA-S exhibited a smaller charge-transfer resistance ($R_{ct}$) compared with that of C-MEA-CCS, resulting in better electrochemical reaction kinetics. When a low-concentration KOH electrolyte (0.5–10 wt%) was used, the all-in-one MEA-S showed a

notably superior performance compared with C-MEA-CCS comprising the Zirfon Thin and $m$-PBI membranes (Supplementary Fig. 38). The alkaline water electrolysis performance of the all-in-one MEA-S and the AEM water electrolysis (AEMWE) performance of C-MEAs using the Sustainion X37-50 membrane were also compared (Fig. 4c). When 5 wt% KOH electrolyte was used, the all-in-one MEA-S showed a superior performance (1.45 A cm⁻² at 1.8 V) compared with the C-MEA-CCM (CoNiS//X37-50//CoNiS), C-MEA-CCM (Pt/C//X37-50//IrO₂) and C-MEA-CCS (CoNiS//X37-50//CoNiS) (0.52, 0.50 and 0.41 A cm⁻² at 1.8 V, respectively). Furthermore, the EIS measurements of MEAs were compared (Fig. 4e). The all-in-one MEA-S exhibited lower $R_{ohm}$ and $R_{ct}$ values than C-MEAs in 5 wt% KOH. The low cell resistances of the all-in-one MEA-S stemmed from the integrated CL/membrane interfaces and well-ordered OH⁻ transfer channel in the membrane. These results indicated that alkaline water electrolysis using the all-in-one MEA was more efficient than ALWE and AEMWE using C-MEAs (Supplementary Table 2). Notably, the performance of the all-in-one MEA-S in 0.5 wt% KOH was better than that of C-MEAs with the Sustainion X37-50 membrane (Supplementary Fig. 39). Alkaline water electrolysis employing the all-in-one MEA-S outperformed most previously reported ALWE[4,36–41] and AEMWE without platinum group metals (PGMs)[14,21,42–47] (Fig. 4f and Supplementary Table 3). Importantly, the performance of the all-in-one MEA-S in 0.5 wt% KOH was comparable to the AEMWE performance in 0.5 wt% KOH and pure water (Supplementary Fig. 40 and

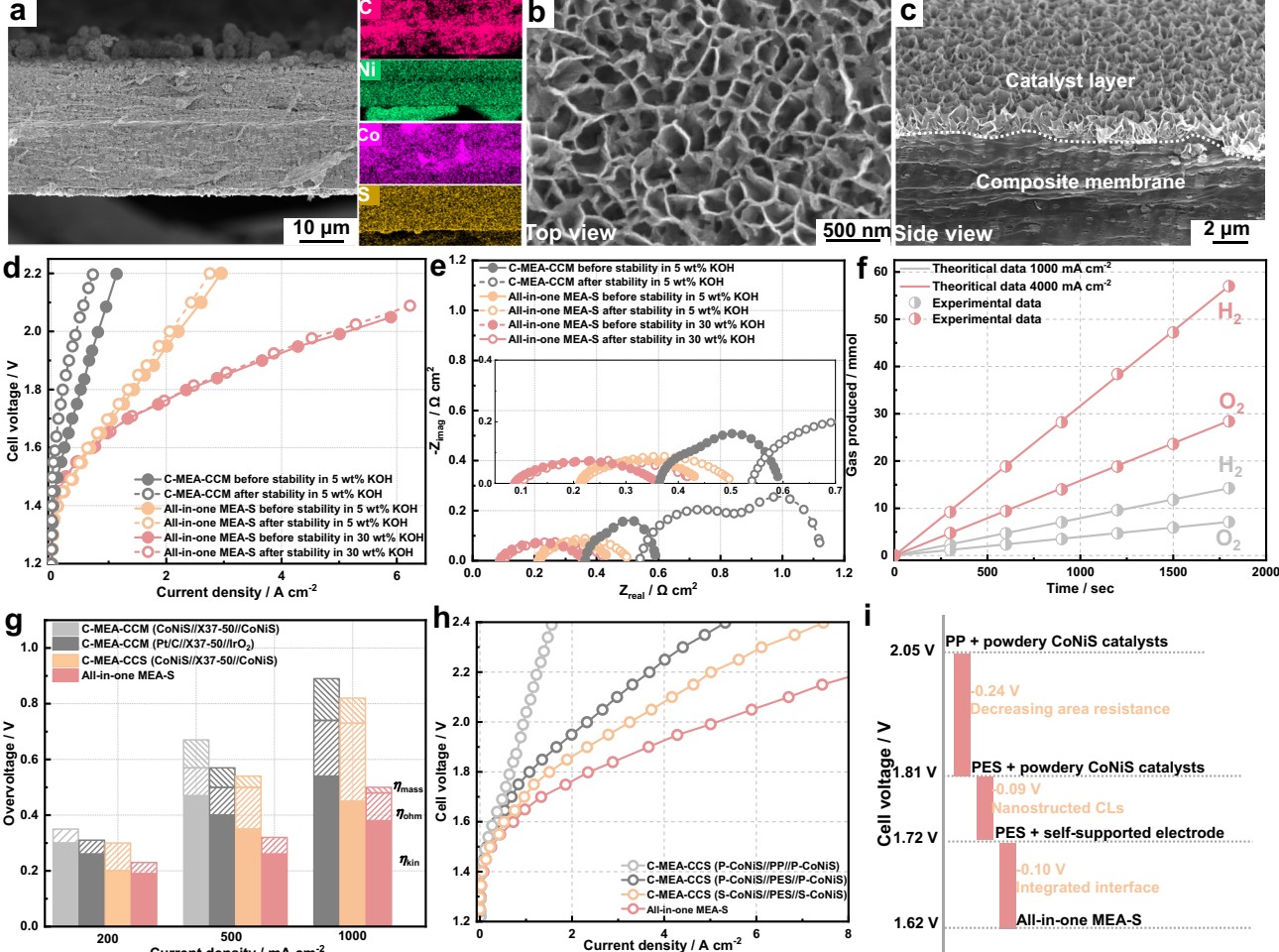

**Fig. 5 | Stability and increased performance of the all-in-one MEAs.** Morphology structure analysis after electrolysis for 600 h: **a** Cross-sectional SEM image of the all-in-one MEA-S and corresponding EDS mapping. **b** Surface and **c** cross-sectional SEM images of the CL. **d** Polarisation curves of the C-MEA-CCM and all-in-one MEA-S before and after stability tests. **e** Nyquist impedance plots of the C-MEA-CCM and all-in-one MEA-S before and after stability tests. **f** Volumes of $H_2$ and $O_2$ generated by the advanced alkaline water electrolysis using the all-in-one MEA-S at different current densities of 1000 and 4000 mA cm⁻² under atmospheric pressure. **g** Bar diagram dividing overpotential into $\eta_{ohm}$, $\eta_{kin}$ and $\eta_{mass}$ at different current densities of 200, 500 and 1000 mA cm⁻². **h** Polarisation curves of the all-in-one MEA-S and controls. The C-MEA-CCS (P-CoNiS//PP//P-CoNiS) and C-MEA-CCS (P-CoNiS//PES//P-CoNiS) were prepared by loading powder CoNiS catalysts on a liquid/gas diffusion layer. The C-MEA-CCS (S-CoNiS//PES//S-CoNiS) was prepared using self-supported CoNiS electrodes. **i** Voltage reductions and their origins. Waterfall plot showing the voltage reductions observed at 1 A cm⁻² (60 °C) and their sources.

Supplementary Table 4). Furthermore, our alkaline water electrolysis showed a similar or higher performance compared with the PEMWE reported in the literature[48–53].

The durability of alkaline water electrolysis based on the all-in-one MEA-S was evaluated at different current densities of 1000, 2000 and 4000 mA cm⁻². As shown in Fig. 4g, the all-in-one MEA-S was operated at 1000 mA cm⁻² in 5 and 30 wt% KOH at 60 °C for over 600 h. The cell voltage of the all-in-one MEA-S slowly increased at rates of 133.3 and 66.7 µV h⁻¹ in 5 and 30 wt% KOH, respectively. In contrast, the cell voltage of the C-MEA-CCM (Pt/C//X37-50//IrO₂) sharply increased after 80 h, and the voltage degradation rate was 1002 µV h⁻¹. Notably, the all-in-one MEA-S also showed durable performance at high current densities of 2000 and 4000 mA cm⁻² for 100 h (Supplementary Fig. 41). The cell voltage of the all-in-one MEA-S was also stable for more than 800 h at 1000 mA cm⁻² in 0.5 wt% KOH. Figure 5a shows the cross-sectional morphology of the all-in-one MEA-S after operation at 1000 mA cm⁻² for 600 h. The CL and membrane structure, and chemical composition of the all-in-one MEA-S were well preserved. Polarisation curves of the all-in-one MEA-S and C-MEA-CCM were recorded after the durability tests (Fig. 5d). The current density of the all-in-one MEA-S decreased by

6.7% and 3.2% at 1.8 V after 600 h of operation in 5 and 30 wt% KOH, respectively, while that of the C-MEA-CCM decreased by 62%. These results suggested the superior stability of the all-in-one MEA-S. Notably, the all-in-one MEA-S still achieved high current densities of 1950 and 4900 mA cm⁻² at 2.0 V in 5 and 30 wt% KOH, respectively. As shown in Fig. 5e, the $R_{ohm}$ value of the C-MEA-CCM increased from 360 to 540 mΩ cm². In contrast, the $R_{ohm}$ value of the all-in-one MEA-S operating in 5 and 10 wt% KOH showed no obvious increase. To assess the gas permeability of the all-in-one MEAs, anodic hydrogen contents were detected by gas chromatography. The all-in-one MEA-S produced hydrogen contents of <0.20 vol% over the current density range of 50–1000 mA cm⁻², which was notably lower than the hydrogen crossover rates reported for conventional separators (Supplementary Fig. 42 and Supplementary Table 6). Furthermore, advanced alkaline water electrolysis using the all-in-one MEA-S produced $H_2$ with a purity of 99.981% at a high current density of 1000 mA cm⁻² (Supplementary Fig. 43). To verify if all electrons were utilised in water electrolysis, the volumes of $H_2$ and $O_2$ produced by the all-in-one MEA-S were measured at different current densities of 1000 and 4000 mA cm⁻² (Fig. 5f), showing almost 100% Faradaic efficiency.

To explain the increased efficiency of the all-in-one MEA, analysis was performed to divide the overvoltage into ohmic overpotential ($\eta_{ohm}$), kinetic overpotential ($\eta_{kin}$) and mass-transfer overpotential ($\eta_{mass}$). The all-in-one MEA-S exhibited significantly lower $\eta_{ohm}$ and $\eta_{kin}$ values compared with the C-MEA-CCM and C-MEA-CCS at low current density (200 mA cm$^{-2}$) owing to the high electrocatalytic activity of CoNiS and the low area resistance of the composite membrane. As the current density increased, the $\eta_{ohm}$ value increased at a constant ratio for all MEAs. When the current density increased from 500 to 1000 mA cm$^{-2}$, the $\eta_{mass}/\eta_{total}$ ratio of the C-MEA-CCS (CoNiS//X37-50//CoNiS) significantly increased to 12.2%. In sharp contrast, the $\eta_{mass}/\eta_{total}$ ratio of the all-in-one MEA-S only increased to 7.8%. The 3D-ordered CL with a microstructured surface allowed instant absorption of the KOH electrolyte and quick release of O$_2$ gas bubbles, resulting in a low $\eta_{mass}$ value and high water electrolysis activity. As shown in Supplementary Fig. 44, the droplets of KOH electrolyte spread immediately when coming into contact with the CL surface of the all-in-one MEA-S. Furthermore, the highly porous CL promoted the departure of O$_2$ bubbles from the CL surface, whereas O$_2$ bubbles readily covered the conventional CL. The high alkaline water electrolysis performance of the all-in-one MEA-S might be attributed to the high bifunctional activity of the CoNiS nanosheet arrays, low area resistance of the composite membrane and integrated CL/membrane interface.

To elucidate the major factors contributing to the increased performance of the all-in-one MEA-S, an alkaline electrolyser with a pristine PP porous membrane and powder CoNiS catalyst was used as a baseline (Fig. 5h). The cell required 2.0 V to obtain a current density of 1 A cm$^{-2}$. When the pristine PP porous membrane was replaced with a polyether sulfone (PES) porous membrane, the voltage required for 1 A cm$^{-2}$ was 0.24 V less, at 1.76 V. The hydrophilic PES membranes had an average pore diameter of 5 μm (Supplementary Fig. 45). The PES membrane displayed a low area resistance of 70 mΩ cm$^2$ owing to its high porosity, which was significantly lower than that of the pristine PP porous membrane and comparable to that of the composite membrane in the all-in-one MEA-S. This decreased cell voltage was almost entirely due to the low area resistance of the membrane. To determine the impact of the 3D-ordered CLs on the electrolysis performance, the powder CoNiS catalysts were replaced with self-supported electrodes with CoNiS nanosheet arrays (Supplementary Fig. 46). The alkaline electrolyser with the PES separator and self-supported electrode required 1.67 V to achieve 1 A cm$^{-2}$. This voltage decrease of 0.09 V was attributed to the binder-free, porous, ordered CL structure. For 3D-ordered CL directly grown on the surface of the hydrophilic PP composite membrane, namely, the all-in-one MEA-S, advanced alkaline water electrolysis required only 1.57 V to generate 1 A cm$^{-2}$. This constituted a further decrease of 0.1 V owing to the integrated CL/membrane interfaces accelerating interfacial mass transfer.

The construction of intergrown CL on low-cost porous membranes is industrially attractive for cost-effective hydrogen production. The CoNiS nanosheet arrays can also be formed on a porous Zirfon Thin membrane, forming an all-in-one MEA-Z (Supplementary Fig. 47). To improve the mechanical properties of all-in-one MEAs, we selected a commercial polyethylene-reinforced polytetrafluoroethylene (PE-reinforced PTFE) porous membrane as a substrate for preparing a large-area all-in-one MEA-R. As shown in Fig. 6a, the all-in-one MEA-R with a size of 30 × 40 cm$^2$ was successfully prepared, and exhibited 3D-ordered nanoflower arrays on the surface of PE nanofibres and PTFE (Supplementary Fig. 48). To further confirm the generality of the CL intergrowth strategy, CoNiS nanowire arrays were successfully constructed on the PP membrane surface, forming an all-in-one MEA-NW. The CoNiS nanowire arrays were mainly dominated by needle-like nanostructure arrays several micrometres in length (Supplementary Fig. 49). The 3D-ordered nanostructure exhibited a superhigh degree of active site exposure and high-efficiency mass transfer paths[54,55]. Notably, we also constructed CLs with a hierarchical architecture using the oriented

intergrowth strategy, demonstrating a 3D hierarchical core-shell structure of CoNiS@FeNi LDH on a PP membrane surface (Supplementary Fig. 50). The polarisation curves of the all-in-one MEA-R, all-in-one MEA-NW and all-in-one MEA (CoNiS@FeNi LDH) are shown in Fig. 6b, with a current density of 1000 mA cm$^{-2}$ in 30 wt% KOH achieved at 1.78, 1.70 and 1.57 V, respectively. The all-in-one MEA-R exhibited a lower current density than the all-in-one MEA-S owing to the high area resistance (200 mΩ cm$^2$) of the reinforced composite membrane. In spite of this, the electrolysis performance of all-in-one MEA-R was comparable to those of the state-of-the-art AEMWE and PEMWE, and was sufficient for advanced alkaline water electrolysis applications. In contrast, the all-in-one MEA-NW and all-in-one MEA (CoNiS@FeNi LDH) exhibited lower voltages than the all-in-one MEA-S owing to the high bifunctional activity of the CLs. The in-situ stabilities of the all-in-one MEA-R and all-in-one MEA (CoNiS@FeNi LDH) were evaluated at 60 °C. As shown in Fig. 6d, the all-in-one MEA-R with an initial cell voltage of ~2.0 V operated stably at a high current density of 2000 mA cm$^{-2}$ for ~1200 h without any voltage decay owing to its preferable mechanical properties[4]. Impressively, the all-in-one MEA (CoNiS@FeNi LDH) demonstrated a low voltage of 1.57 V at a current density of 1000 mA cm$^{-2}$ for 1000 h, which represented a cell energy efficiency of 94% (HHV). To our knowledge, this cell performance at a high current density of 1000 mA cm$^{-2}$ is among the best reported competitive alkaline water electrolysis performance (Fig. 6e and Supplementary Table 7).

A key strategy for reducing the levelized cost of hydrogen (LCOH) involves improving the energy efficiency of alkaline water electrolysers. Figure 6c shows a comparison of the LCOH of electrolyser stacks based on the C-MEA-CCS (CoNiS//Zirfon Thin//CoNiS), C-MEA-CCM (Pt/C//X37-50//IrO$_2$) and all-in-one MEA-S. The LCOH of the all-in-one MEA-S was lower than that of C-MEAs in categories including electricity, maintenance and capital expenditure costs. Therefore, the all-in-one MEAs show potential to improve the competitiveness of green hydrogen against fossil fuels.

## Discussion

The de novo design of all-in-one MEAs has been exploited for advanced alkaline water electrolysis. Through the oriented intergrowth of nanosheet arrays from a porous membrane, 3D-ordered CLs were constructed inside the porous membranes, building transport highways to facilitate gas/liquid and ion mobility. The low area resistance and high bifunctional activity of all-in-one MEAs were characterised in detail. As a proof of concept, advanced alkaline water electrolysis using this all-in-one MEA exhibited a high current density of 1000 mA cm$^{-2}$ at 1.57 V because the 3D-ordered transport highways in the membrane and CLs effectively transferred OH$^-$ while promoting gas/liquid transport. More importantly, advanced alkaline water electrolysis using a reinforced all-in-one MEA maintained a current density of 2000 mA cm$^{-2}$ for over 1000 h. These findings provide an efficient approach for developing next-generation MEAs for advanced alkaline water electrolysis.

## Methods

### Materials

Ferrous sulfate (FeSO$_4$·7H$_2$O, 99%), cobalt nitrate (Co(NO$_3$)$_2$·6H$_2$O, 99%), nickel nitrate (Ni(NO$_3$)$_2$·6H$_2$O, 99%), urea (CO(NH$_2$)$_2$, 99%), sodium sulfide (Na$_2$S·9H$_2$O, 99%) and potassium hydroxide (KOH, 99.99%) were purchased from Sigma Aldrich and used as received. PP (Celgard 2400, 20 μm) and PE separators (20 μm) were purchased from Suzhou Sinero Technology Co., Ltd. (Suzhou, China). PE-reinforced PTFE (~100 μm) porous membrane was obtained from Changzhou Jinchun. Co., Ltd. (Changzhou, China). Zirfon Thin (~200 μm) and Zirfon PERL (500 μm) separators were provided by Agfa. Fumasep FAA-3-20 (20 μm), FAA-3-50 (50 μm) and *m*-PBI membranes (20 μm) were purchased from Fumatech. Sustainion X37-50

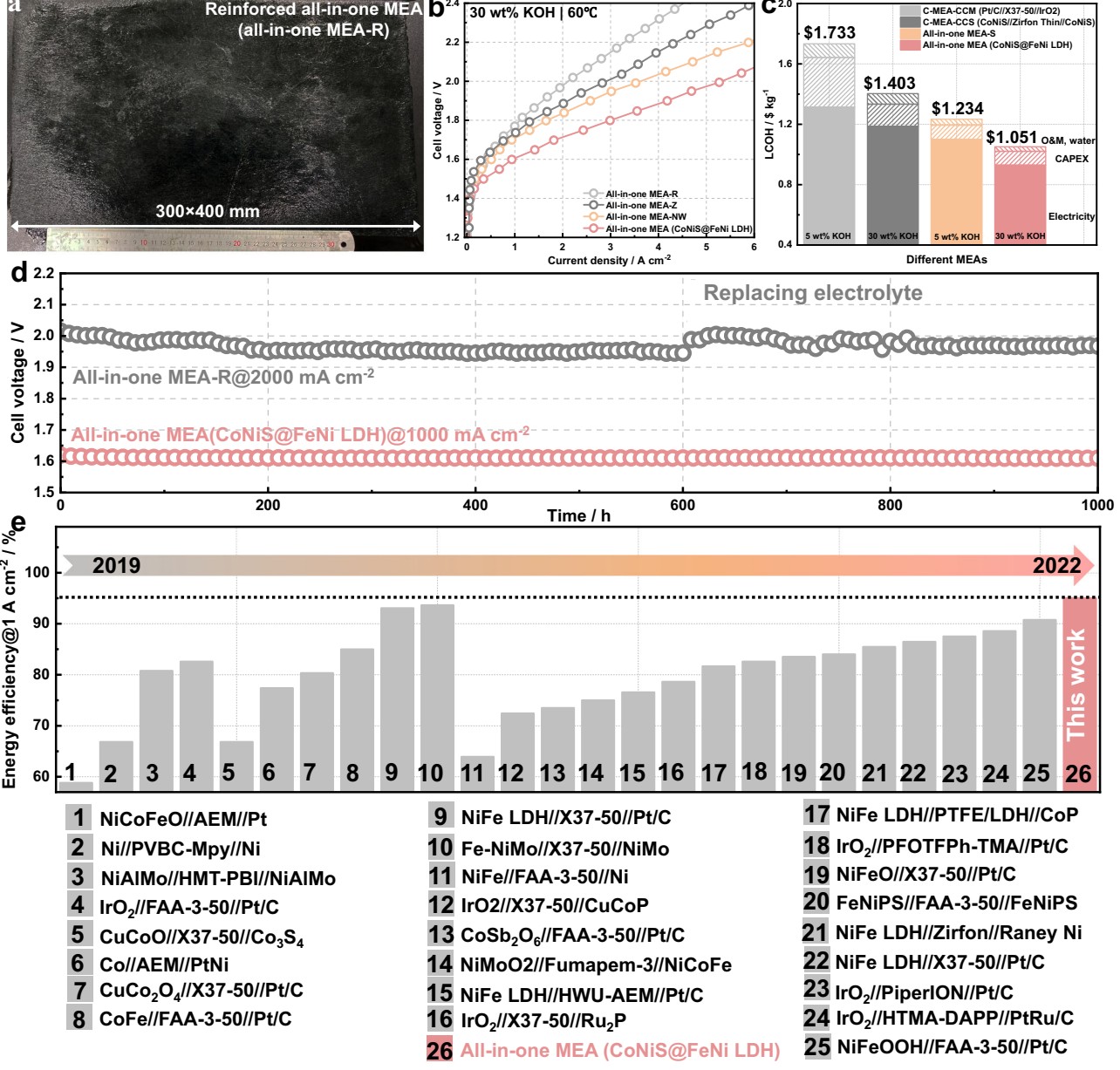

**Fig. 6 | Generality of the oriented CL intergrowth strategy. a** Photograph of the all-in-one MEA-R prepared. **b** Polarisation curves of the all-in-one MEA-R, all-in-one MEA-N and all-in-one MEA (CoNiS@FeNi LDH) in 30 wt% KOH at 60 °C. **c** LCOH of advanced alkaline water electrolysis using the C-MEA-CCM (Pt/C//X37-50//IrO₂), C-MEA-CCS (CoNiS//Zirfon Thin//CoNiS) and all-in-one MEA-S. **d** Durability cell voltage-time plots for the all-in-one MEA-R and all-in-one MEA (CoNiS@FeNi LDH) at constant current density in 30 wt% KOH at 60 °C. **e** Energy efficiency comparison of the all-in-one MEA (CoNiS@FeNi LDH) with reported alkaline water electrolysis at 1 A cm⁻² (Supplementary Table 7).

membranes (50 μm) were purchased from Dioxide Materials, Inc. Ni foil and Ni foam were purchased from Kunshan GuangJiaYuan New Materials Co. Ltd. (Kunshan, China). The water used in this study was 18 MΩ Milli-Q deionized (DI) water.

### Preparation of the all-in-one MEA-L

The PP porous separator, cut to a suitable size ($2 \times 4$ cm²), was first treated with ethanol solution at 80 °C for 2 h. This treatment removed surface impurities and improved surface hydrophilicity. $Co(NO_3)_2 \cdot 6H_2O$ (700 mg), $Ni(NO_3)_2 \cdot 6H_2O$ (500 mg) and urea (900 mg) were dissolved in ethanol (70 mL) to obtain a well-dispersed solution. This was transferred to a 100-mL Teflon-lined stainless autoclave, and treated PP porous separators were then immersed in the solution. To promote the complete infiltration of precursors into the porous separators, the autoclave was placed in an ultrasonic cleaner for 10 min. The autoclave was then sealed and placed in an oven at 120 °C for 6 h. After the reaction, the autoclave was cooled to room temperature, and the obtained products were rinsed with DI water and dried at room temperature.

### Preparation of the all-in-one MEA-S

To obtain the all-in-one MEA-S, the as-prepared all-in-one MEA-L was subjected to sulfurization. $Na_2S \cdot 9H_2O$ (1.0 g) was completely dissolved in DI water (80 mL), and then transferred to a 100-mL autoclave containing the all-in-one MEA-L, and the autoclave was then heated to 120 °C for 6 h. After reaction completion, the obtained all-in-one MEA-S was washed with DI water and dried at room temperature.

## Preparation of the all-in-one MEA-Z

The synthetic route for the all-in-one MEA-Z followed the same preparation steps used to obtain the all-in-one MEA-S, but the PP porous membrane was replaced with a commercial Zirfon Thin membrane.

## Preparation of the all-in-one MEA-R

The all-in-one MEA-R was prepared using a PE-reinforced PTFE porous membrane with a pore size of 0.1 μm, high porosity of >90%, and thickness of 100 μm. The synthetic route for the all-in-one MEA-R followed the same preparation steps used to obtain the all-in-one MEA-S, but the PP porous membrane was replaced with a PE-reinforced PTFE porous membrane.

## Preparation of the all-in-one MEA-NS

The synthetic route for the all-in-one MEA-NS followed the same preparation steps used to obtain all-in-one MEA-S, but $Co(NO_3)_2.6H_2O$ and $Ni(NO_3)_2.6H_2O$ were replaced with $CoCl_2.6H_2O$ and $NiCl_2.6H_2O$.

## Preparation of the all-in-one MEA (CoNiS@FeNi LDH)

For the preparation of the all-in-one MEA (CoNiS@FeNi LDH), the precursor solution was prepared by mixing $Ni(NO_3)_2·6H_2O$ (1.6 mmol), $FeSO_4·7H_2O$ (0.4 mmol), urea (8 mmol) and $H_2O$ (60 mL). Then, the precursor solution containing the as-prepared all-in-one MEA-S was treated by a hydrothermal method at 100 °C for 3 h.

## Preparation of conventional MEAs

C-MEAs were prepared using different fabrication methods, including CCS and CCM. For C-MEA-CCM preparation, commercial 40% Pt/C (Johnson Matthey) and $IrO_2$ (Alfa Aesar, MA, USA) catalysts were employed as HER and OER electrocatalysts, respectively. The cathode and anode inks were prepared according to recent literature[14]. For anode ink preparation, $IrO_2$ powder (5 mg) was mixed with DI water (50 mg), isopropyl alcohol (100 mg) and quaternary ammonia poly(*N*-methyl-piperidine-co-p-terphenyl) (QAPPT) ionomer (5.5 mg; 10 wt%). The anode ionomer ratio was 10 wt%. For cathode ink preparation, Pt/C powder (5 mg) was mixed with DI water (25 mg), isopropyl alcohol (50 mg) and QAPPT ionomer (12.5 mg; 10 wt%). The cathode ionomer ratio was 20 wt%. The dispersed inks were sprayed onto both sides of an AEM (Sustainion X37-50). The $IrO_2$ and Pt/C loadings were 2.0 and 0.5 mg cm$^{-2}$, respectively. The cathode carbon paper and anode Ni foam were sandwiched with the catalyst coated membrane and pressed to form a MEA at 70 °C for 3 min at 0.1 MPa. C-MEAs-CCS with powder catalysts were prepared using the same catalyst ink and loading steps as those performed for the C-MEA-CCM. The dispersed ink was sprayed on Ni foam and carbon paper, forming the anode and cathode, respectively. The as-prepared electrodes were pressed with a membrane (including PP, PES and Sustainion X37-50 membranes) at 70 °C for 3 min at 0.1 MPa. To prepare the C-MEA-CCS with self-supported electrodes, the synthesis process of the self-supported CoNiS electrode was the same as that used for all-in-one MEA-S, but the porous PP membrane was replaced with Ni foam or carbon paper. The as-prepared self-supported electrodes were pressed with a membrane (such as PES, Zirfon Thin, *m*-PBI and Sustainion X37-50) at 70 °C for 3 min at 0.1 MPa. The cathode and anode CL loadings using powder CoNiS catalysts on the membrane or liquid/gas diffusion layer were both 2.5 mg cm$^{-2}$.

## Physical characterisation

The micromorphologies and nanostructures of the all-in-one MEAs were analysed by field-emission SEM (FE-SEM, Merlin) and TEM (JEM2010, Japan). To observe the CL/membrane interface, a TESCAN S9000 SEM instrument equipped with focused ion beam (FIB) accessories was used. The FIB was used for surface etching and further in-depth elemental analysis. The surface structures of the CLs were analysed by means of a white light interferometry profiling system (Bruker Contour GT K0 G, Germany). AFM (MFP-3D-SA, USA) was utilised to determine the CoNiS nanosheet thickness. Crystallographic information for CoNi LDH and CoNiS was obtained by XRD using Cu Kα irradiation ($\lambda = 1.5406$ Å). Compositional information was obtained by XPS (Ulvac-Phi, Japan) using a Thermo Scientific K-Alpha spectrometer. X-ray micro-computed tomography (Xradia 620 Versa) was used to obtain 3D structures of the samples. Nitrogen adsorption-desorption isotherms were obtained with an Autosorb-IQ2-MP-C system at 77 K. The specific surface area was evaluated following the multipoint Brunauer-Emmett-Teller (BET) method. The average pore diameters and porosities of the separators and membranes were determined using MIP (AutoPore Iv 9510) and CFP (PMI, USA). The interfacial bonding strength of the MEAs was measured using an electronic universal material testing machine (WDW-100). The contact angles were measured by a contact angle tester (Data physics Co. Ltd., GER OCA-20). More physical characterisation of the membranes can be found in the Supplementary Materials.

## Electrochemical characterisation

To evaluate the electrocatalytic activity of the catalyst layers, catalyst layers were prepared on Ni foil using the same method. The catalyst loading was 2.5 mg cm$^{-2}$. Commercial $IrO_2$ and Pt/C were also sprayed on the Ni foil. The $IrO_2$ and Pt/C loadings were 2.0 and 0.5 mg cm$^{-2}$, respectively. The OER and HER activities of the catalyst layers were evaluated in a standard three-electrode system, in which graphite plate (1 cm × 1 cm), the as-synthesised electrode (1 cm × 1 cm) and Hg/HgO were directly utilised as the counter electrode, working electrode and reference electrode, respectivity, in a 1 M KOH solution. The OER polarisation curves were determined by LSV in the potential range of 1.1–1.8 V vs. RHE at a scan rate of 2 mV s$^{-1}$. The HER polarisation curves were determined by LSV in the potential range −0.1 to −0.6 V vs. RHE at a scan rate of 2 mV s$^{-1}$. EIS was performed over a frequency range from 0.1 to 100 Hz and an amplitude of 10 mV. Moreover, the ohmic resistance was measured by EIS to compensate for the iR loss. The OER and HER durability evolution of the as-prepared electrodes was studied by performing galvanostatic measurements at different current densities of 100 and 500 mA cm$^{-2}$ for 100 h. Moreover, the polarisation curves were recorded after the stability tests.

## Advanced alkaline water electrolysis

The electrolysis performance of the MEAs was assessed with a home-made alkaline water electrolyser with an active area of 5 cm$^2$. Ni foam and carbon paper were used as the anode and cathode liquid/gas diffusion layers, respectively. The cathode carbon paper and anode Ni foam were sandwiched with the as-prepared all-in-one MEA and pressed at 0.1 MPa and 70 °C for 3 min to form a five-layered MEA. Electrochemical measurements of alkaline electrolysis were performed using a Wuhan LAND electrochemical station. Polarisation curves without iR compensation were obtained by recording the current density in the voltage range of 1.1–2.4 V at 60 °C. EIS measurements were performed using an electrochemical workstation (VersaSTAT-3F, Princeton Applied Research). The frequency range was 10 kHz to 0.01 Hz, and the amplitude was 5 mV. Several electrolyte concentrations (0.5, 5, 10 and 30 wt% KOH) were used in both the anode and cathode compartments at a flow rate of 50 mL min$^{-1}$. Faradaic efficiency was calculated by measuring volumes of gases produced in alkaline water electrolysis at current densities of 1000 and 4000 mA cm$^{-2}$.

## Data availability

Data are available from the corresponding author upon request. Source data are provided with this paper.

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

## Acknowledgements

This work was supported by the National Key R&D Program of China (2020YFB1505602 and 2018YFE0202001) and the National Natural Science Foundation of China (21776154).

## Author contributions

L.W. discussed data and wrote the original draft. M.P., J.L. and Z.X. discussed the results. H.Z. and Q.X. analysed cell data. B.W. reviewed this manuscript. All of the authors discussed the data on the paper.

## Competing interests

The authors declare no competing interests.
