## [Peer Review File · Nature Communications]

Oriented intergrowth of the catalyst layer in membrane electrode assembly for alkaline water electrolysisREVIEWER COMMENTS

Reviewer #1 (Remarks to the Author):

This manuscript deals an all-in-one MEA with 3D integrated CL/membrane interfaces by using CoNi LDH (or CoNiS) and porous PP membrane. The reviewer is very interested all-in one MEA and it's results for alkaline water electrolysis, but the reviewer can't be sure the paper is suitable for publication in Nature communications in this present state. Few changes are needed to improve the manuscript:

1. Although the author expressed "Biomimetic" or "inspired by vertical growth of a tree on ground", it is difficult to find a correlation with the developed all-in-one MEA.
2. The water electrolysis electrode and the fuel cell electrode are different. For examples, in the case of commercially available alkaline water electrolysis based on zirconium membrane, a zero-gap exists between the electrode and the membrane to block the electrical conductivity between the electrodes. Thus, in this paper, manufacturing all-in-one MEA to reduce the interfacial resistance between the electrode and the membrane is not considered an important part of water electrolysis.
3. Also, it seems that the author should be clear whether the application of this development MEA is AEMWE or development for AEW. The MEA manufacturing methods and development directions in the two fields are different. Reviewers are confused by the presented data by the authors.
4. For example, m-PBI, FAA-3, and zirconium were compared, but the explanation of how MEA was fabrication (What kind of ionomer was used for each) and compared is insufficient. What is the cell temperature? What is the thickness of the manufactured all in one MEA? Have you considered the performance change according to the thickness of m-PBI, FAA, and zirconium used? m-PBI and FAA used are high-density polymer electrolyte membranes and zirconium is a diaphragm. Is an integral MEA a porous diaphragm? Or is it a kind of polymer electrolyte membrane that gas does not pass through? In addition, Why didn't you compare the sustain membrane of Dioxide Materials, which is known to have the best performance in AEMWE?
5. Caption of Figure and scheme with SI have to be modified according to the manuscript. For example, in Figure 1d there is no lamellar structure. This can be seen in 1e. Figure S11 also does not match the text. Even Figure S30 and S31 have the same caption.
6. Experimental Section also have to be modified. More detailed explanation needed. For example, The sample names for the case of using pp porous membrane and the case of using PE-reinforced PTFE are used interchangeably. Also what is MEA-R, CoNiS NW etc? In addition, there is no detailed description of the MEA manufacturing method in these cases, and the description of each analysis and experimental method (Ex. Interfacial bonding strength, porosity of all in one MEA) is insufficient. With the current experimental method written by the author, it is difficult to reproduce.
7. CoNi-LDH material is a very popular material in supercapacitors under alkaline conditions. It is also used as an amphoteric catalyst in alkaline water electrolysis. Also, CoNi-LDH material has been grown on various supports (ZIF, Porous polymer substrate, etc) and used as capacitors or catalysts. Therefore, it seems that the novelty of this study needs more explanation. Is this the first time this material has been applied to alkaline water electrolysis?
8. After sulfurination (CoNi-LDH to CoNiS), There is no evidence of more roughness.(Figure 1c)
9. The reviewer is really curious, in the case of AEW and AEMWE, the diaphragm or the electrolyte membrane plays a role in blocking the electrical conductivity between the two electrodes, respectively. However, in the all-in one MEA manufactured by the author, there is no diaphragm or zero gap to block electrical conductivity. The current all-in-one MEA is considered a short circuit system. It was understood that CoNi-LDH or CoNiS material acts as an amphoteric catalyst and shows high activity for HER and OER, showing high performance in water electrolysis. However, how can the electrical conductivity between the two electrodes be blocked in the current system? Where is the electrolyte membrane (or diaphragm) in the system now, and where are the electrodes? Where does each electrode reaction take place? A detailed explanation is needed for this. In Scheme 1b and c, the CoNi-LDH or CoNiS material filled with the porous PP membrane, as if it were a different material from the electrode, was displayed in different colors. However, it is correct to express it in the same color as the electrode.

10. If the manufactured three-dimensionally aligned CoNi-LDH or CoNiS electrode shows good performance, how about forming this electrode separately and evaluating the MEA performance compared to the all-in-one MEA?
11. The evidence is needed for the levelized cost of hydrogen (LCOH) and the calculated energy efficiency.
12. Manuscript needs substantial English polishing and typos revision.

Reviewer #2 (Remarks to the Author):

The authors of 'Biomimetic approach to the oriented intergrowth of catalyst layer in membrane electrode assembly for alkaline water electrolyzer' presented a study in which the bifunctionality of their intergrown 3-D catalyst layers with advanced performance for alkaline water electrolysis was showcased. The authors made a step-by-step representation of their catalyst (CoNi LDH \diamond CoNiS) and growth onto PP porous membrane later replaced by PE reinforced PTFE membrane and use of CoNiS@FeNi LDH as catalyst. The paper carries value as the fusing of electrode and membrane should bring about a decrease in interfacial resistances and improved performances beyond that of CCM design. It would be of interest to see the performance of such an all-in-one MEA design also reported under pressure operation in future works for AEMWE. There are comments and the appeal to authors to have the manuscript submitted for language editing. There are typos, untypical word choices and incorrect use of prepositions (as example) present throughout the manuscript and SI. I support the publication of manuscript after major revision.

Abstract

21: add the KOH concentration the measurement was performed at here. It is relevant when interpreting AWE or AEMWE performances. Cell potential recorded at 1 A/cm² in 30 wt% KOH.

Introduction

29-30: 'A porous membrane permeable to hydroxide ions prevents gas permeability'. – Disagree with the use of 'prevents' in this context – should rather be limits or decreases gas permeability and hence mixing of O₂ and H₂ product gases. H₂ crossover still occurs during operation.

45: 'fall off' – rather to refer to detachment of catalyst here.

50: reference 12 refers mainly to PEMs. Consider to add references for AEM studies conducted on CCM methods.

Results

Scheme 1: The arrows for OH⁻ directed from anode in (a) and (b) are misleading. OH⁻ species will be conducted from cathode to the anode.

65: add KOH concentration operated in here as well.

102: Add reference in text to the methods section where more information is shared on the conventional CCM. (refer to 425) does not seem to be the optimal catalyst ink formulation for well performing MEA. Which ionomer was used? No water added to the solvent mixture? Also include thickness of membrane here.

108: ref 20 & 21. Worthy references, but it should be kept in mind the OH⁻ transport mechanism for AEM electrolyzers (liquid&gas phase) does pose different challenges to that of PEM fuel cells (gas phase).

142: There is mention of thickness of the catalyst growth, but is there any indication (estimated) of the loading for anode and cathode for the all-in-one membranes? Or which is later then electrochemically compared with other OER and HER catalysts (Fig 1 I & j) in terms of LSV an CA measurements?

199-201: It seems the use of increased and decreased area resistance (or 30 and 10 wt% KOH exchanged in the sentence) in the sentence should be swapped . Please read again.

155: Was it tried to grow the 3D-ordered CLs onto porous diaphragm Zirfon besides from the AEM (FAA-3-20) and m-PBI mentioned in the study?

176: There is mention of earth abundant cobalt, however the mining conditions and exploiting of the workers deserve consideration in this context while it is anticipated that the battery market will increase their cobalt demands in future, driving up the price and scarcity of cobalt.

230: Here the mention of including Zirfon in membrane characterization. Could the authors add the porosity /pore size of the material in comparison to porous PP used?

Fig 3 (e)/ line 270: Are the references for these SoA data reported on AEM and PEMWE results listed in the SI? It could be required to add the references for the presented studies in this graph. For example are the AEMWE data points reported for non-PGM electrode studies only, or is AEMWE performances for KOH electrolyte supported studies only reported? Recent works in pure water have also proved promising, such as <https://doi.org/10.1038/s41560-020-0577-x> and <https://doi.org/10.1021/acscatal.0c04200>.

278: Is the 30wt% KOH impedance data also available for the all in-one MEA? Why not provide both 10 and 30 wt% data sets? Not able to accurately compare impedance spectra reported after 600h stability measured in 30 wt% KOH (Fig 4 e).

298: The authors mention gas permeability measurements, and since 100% Faradaic efficiency is reported at both 1000 and 4000 A/ cm² do the authors assume there is no hydrogen crossover on the anode side? This is usually only accurately determined by measuring hydrogen crossover at the anode directly.

336: When referring to powdery CoNiS catalyst use with the PP membrane, does the authors refer to spreading of a catalyst ink onto the membrane? The description of preparing these MEAs is missing from the method section. What is the thickness of PES porous membrane used?

337: Please see Fig 4 h(i) and line 339: h (ii) notation. The reference here does not make sense or it is a typo for h & i figures?

340: reference to Fig S40 – please double check the captions of Figures in SI. Fig S40 show images of PTFE supported PP membrane. Inconsistent with reference from main text. There is no SEM images representing PES membrane in the SI.

362: Supplier of the commercially used PE-reinforced PTFE membrane is not mentioned.

Fig 5 (b) is a comparison of different catalyst and porous membrane (PP vs PE reinforced with PTFE) investigated in this study for the purpose of finding low cost solutions. Can this be confirmed that the all-in-one MEA (CoNiS@FeNi LDH) @1 A/cm² is the PP porous membranes used in all-in-one MEA-S? The polarization curve Fig 3 (b) for all-in-one MEA-S 30wt% KOH is very similar to performance of all-in-one MEA (CoNiS@FeNi LDH) in Fig 5 (b), however the cell voltage @1 A/cm² is lower and degradation rate smaller for last mentioned catalyst variation. The MEA (CoNiS@FeNi LDH) preparation method with new catalyst is not included in the method section. Please review.

351: Please specify which all-in-one MEA the Nr 26 in Fig 5 (e) refers from those evaluated in the study.

Method

439: Missing information on assembly: truly zero-gap tested MEA? Fig 3 (a) illustrates the configuration of the cell, but whether the MEA and LGDLs are compressed isn't clear. What is the cell size (active area of the cell) tested?

SI: please review for spelling mistakes and control that figure captions fit the figures content.

Response to reviewers' comments

Manuscript ID: NCOMMS-22-22077

Title: Biomimetic approach to the oriented intergrowth of catalyst layer in membrane electrode assembly for alkaline water electrolyzer

Authors: Lei Wan, Maobin Pang, Junfa Le, Ziang Xu, Hangyu Zhou, Qin Xu and Baoguo Wang*

We have considered reviewers' comments carefully and provided responses point by point to all questions. We also revised the original manuscript to make a new version as updated file, and the main revised part was highlighted with yellow color. The responses to the reviewers' comments are listed as the followings.

To Reviewer 1:

Referee 1:

This manuscript deals an all-in-one MEA with 3D integrated CL/membrane interfaces by using CoNi LDH (or CoNiS) and porous PP membrane. The reviewer is very interested all-in one MEA and it's results for alkaline water electrolysis, but the reviewer can't be sure the paper is suitable for publication in Nature communications in this present state. Few changes are needed to improve the manuscript:

Response to the comments:

We are grateful for the time and effort. Reviewer 1 has reviewed our manuscript, which are constructive and valuable to improve our manuscript.

Comment 1:

Although the author expressed "Biomimetic" or "inspired by vertical growth of a tree on ground", it is difficult to find a correlation with the developed all-in-one MEA.

Response to the comments 1:

Thanks for the constructive comments. The related sentences “Biomimetic” or “inspired by vertical growth of a tree on ground” have been deleted or alternated in the revised manuscript. Moreover, the title has been corrected as “Oriented intergrowth of catalyst layer in membrane electrode assembly for alkaline water electrolyzer”.

Comment 2:

The water electrolysis electrode and the fuel cell electrode are different. For examples, in the case of commercially available alkaline water electrolysis based on zirconium membrane, a zero-gap exists between the electrode and the membrane to block the electrical conductivity between the electrodes. Thus, in this paper, manufacturing all-in-one MEA to reduce the interfacial resistance between the electrode and the membrane is not considered an important part of water electrolysis.

Response to the comments 2:

We really appreciate the reviewer for providing insightful comments on our work. Firstly, we agree with the review’s comments that a zero-gap exists between the electrode and the membrane to block the electrical conductivity between the electrodes. However, we should explain that the zero-gap cell design significantly reduces the ohmic resistance contribution from the electrolyte between the two electrodes (*RSC Adv.*, 2016,6, 100643-100651).

For commercially available alkaline water electrolysis based on Zirfon membrane, the self-supported electrodes and porous diaphragm are widely used. The used concentrated KOH electrolyte (30 wt% KOH) can significantly reduce the ohmic resistance of diaphragm and the interfacial resistance of the gap between self-supported electrode and diaphragm (*ACS Materials Lett.* 2021, 3, 224–234; *ACS Sustainable Chem. Eng.* 2018, 6, 4829–4837). To further investigate the impact of the interfacial resistance on the performance of conventional MEA-CCS (C-MEA-CCS) using Zirfon membrane and self-supported electrodes, we utilized the hot-pressing method to regulate the binding force between self-supported electrodes and Zirfon membrane. As shown in Fig. R1a-b, the introduction of hot pressing will increase the electrolysis performance of MEAs, especially at high current density. Furthermore, to characterize the ohmic resistances in MEAs, EIS was performed at 1.4 V (Fig. R1c). The intercept in the high-frequency region represents the ohmic resistance, which consists of the ionic

and electronic resistances. Improving the hot-pressing pressure reduced the ohmic resistance of cell.

More importantly, we also investigated the impact of the interfacial resistance on the performance of C-MEA-CCS using Sustainion X37-50 membrane and self-supported electrodes. Fig. R2 exhibited the SEM images of the AEM/electrode interface after hot pressing under different pressures. It can be observed that the interface distance decreases as improving the hot-pressing pressure from 0.1 to 1.0 MPa. When 30 wt% KOH solution is used as the electrolyte, MEAs-CCS with different hot-pressing pressure showed discrepant electrolysis performance (Fig. R1d). Typically, the MEA with 1.0 MPa showed cell voltage of 1.61, 1.76, and 2.03 V at different current densities of 500, 1000, and 2000 mA cm⁻², which are lower than that (1.67, 1.84, and 2.16 V) of the MEA with 0.1 MPa (Fig. R1e). Furthermore, to characterize ohmic resistances in MEAs, EIS was performed at 1.4 V (Fig. R1f). Improving hot-pressing pressure also reduced the ohmic resistance of the cell. This result indicated that reducing the AEM/electrode interfacial resistance can improve the electrolysis performance. When 10 wt% KOH solution is used as the electrolyte, the electrolysis performance difference of MEAs-CCS with different hot-pressing pressure is significant (Fig. R1g). Typically, the MEA with 1.0 MPa showed cell voltage of 1.54, 1.68, and 1.92 V at the different current densities of 250, 500, and 1000 mA cm⁻², which are significantly lower than that (1.65, 1.87, and 2.3 V) of the MEA with 0.1 MPa (Fig. R1h). Improving the hot-pressing pressure can significantly reduce the ohmic resistance of the cell, especially at low concentrated KOH solution (Fig. R1i).

Therefore, we consider that the all-in-one MEA with integrated catalyst layer/membrane interfaces can significantly reduce the interfacial resistance at low concentrated KOH in the range of 0.5 wt% (0.10 M) to 5.0 wt% (1.0 M) KOH. Therefore, additional experiments have been conducted as follows:

Fig. R3a includes the polarization curves of C-MEA-CCM (CoNiS//X37-50//CoNiS), C-MEA-CCM (Pt/C//X37-50//IrO₂), C-MEA-CCS (CoNiS//X37-50//CoNiS) and all-in-one MEA-S in 5 wt% KOH at 60 °C. The overall cell performance of all-in-one MEA-S is significantly better than that of C-MEAs. For example, the cell using all-in-one MEA-S achieves a current density of 1.5 A cm⁻² at 1.80 V, which shows an improvement of 0.95, 1.00, and 1.10 A cm⁻² at the same cell voltage compared with the cells using C-MEAs-CCS (CoNiS//X37-50//CoNiS), C-MEA-CCM (Pt/C//X37-50//IrO₂) and C-MEA-CCM (CoNiS//X37-50//CoNiS), respectively. EIS was performed at a current density of

0.5 A cm⁻² to analyze the differences in the performances of C-MEAs and all-in-one MEA-S in 5 wt% KOH (Fig. R3b). All-in-one MEA-S exhibited lower cell ohmic resistance (R_{ohm}) of 0.21 Ω cm², in comparison with that (0.35, 0.35 and 0.40 Ω cm²) of C-MEA-CCM (CoNiS//X37-50//CoNiS), C-MEA-CCM (Pt/C//X37-50//IrO₂) and C-MEA-CCS (CoNiS//X37-50//CoNiS), respectively. The overall overvoltage was divided into ohmic overvoltage (η_{ohm}), kinetic overvoltage (η_{kin}), and mass transfer overvoltage (η_{mass}). As shown in Fig. R3c, all-in-one MEA-S shows lower η_{ohm} , η_{kin} and η_{mass} over different current densities (200, 500, and 1000 mA cm⁻²) compared with C-MEAs in 5 wt% KOH.

More importantly, All-in-one MEA-S still exhibited a high current density of 1.1 A cm⁻² at 1.80 V in 0.5 wt% KOH (Fig. R3d), which significant higher than that (0.15, 0.40 and 0.40 A cm⁻²) of C-MEA-CCM (CoNiS//X37-50//CoNiS), C-MEA-CCM (Pt/C//X37-50//IrO₂) and C-MEA-CCS (CoNiS//X37-50//CoNiS), respectively. As shown in Fig. R3e, EIS analysis showed the R_{ohm} (0.31 Ω cm²) of all-in-one MEA-S is still lower than that (0.39, 0.40 and 0.45 Ω cm²) of C-MEA-CCM (CoNiS//X37-50//CoNiS), C-MEA-CCM (Pt/C//X37-50//IrO₂) and C-MEA-CCS (CoNiS//X37-50//CoNiS), respectively. Overvoltage analysis exhibited that all-in-one MEA-S has lower η_{ohm} , η_{kin} , and η_{mass} over different current densities (200, 500, and 1000 mA cm⁻²) compared with C-MEAs in 0.5 wt% KOH. All-in-one MEA-S exhibited a stable voltage behavior (1.79-1.82 V) at 1000 mA cm⁻² over the 400 h test (Fig. R3g). In contrast, C-MEA-CCM (Pt/C//X37-50//IrO₂) and C-MEA-CCS (CoNiS//X37-50//CoNiS) exhibited high initial cell voltage of 2.20 and 2.19 V. Especially, C-MEA-CCM (Pt/C//X37-50//IrO₂) showed a voltage increase of up to 2.40 V for 90 h.

In conclusion, this all-in-one MEA features 3D-ordered catalyst layers with large surface areas, low tortuosity of pore structure, integrated CL/membrane interfaces, and a well-ordered OH⁻ transfer channel. Due to this design, high electrolysis performance and stability can be achieved even in low-concentrated KOH solutions (0.5-5 wt%).

Fig. R1 | Performance of MEAs using different hot-pressing pressure. C-MEA-CCS (CoNiS//Zirfon Thin//CoNiS) uses commercial Zirfon Thin (200 μm) and self-supported CoNiS as membrane and electrode, respectively; C-MEA-CCS (CoNiS//X37-50//CoNiS) uses commercial Sustainion X37-50 and self-supported CoNiS as membrane and electrode, respectively. There is no ionomer in the cathode and anode. (a) Polarization curves of C-MEA-CCS (CoNiS//Zirfon Thin//CoNiS) prepared by different hot-pressing pressure (0, 0.2, 0.5 and 1.0 MPa) in 30 wt.% KOH at 60 °C. (b) The cell voltage of MEAs with different hot-pressing pressure at various current densities (500, 1000, and 2000 mA cm^{-2}). (c) Nyquist plot of MEAs with different hot-pressing pressure at 1.4 V_{cell} in 30 wt.% KOH at 60 °C. (d) Polarization curves of C-MEA-CCS (CoNiS//X37-50//CoNiS) prepared by different hot-pressing pressure (0, 0.2, 0.5, and 1.0 MPa) in 30 wt.% KOH at 60 °C. (e) The cell voltage of C-MEA-CCS (CoNiS//X37-50//CoNiS) with different hot-pressing pressure at various current densities (500, 1000, and 2000 mA cm^{-2}). (f) Nyquist plot of C-MEA-CCS (CoNiS//X37-50//CoNiS) with different hot-pressing pressure at 1.4 V_{cell} in 30 wt.% KOH at 60 °C. (g)

Polarization curves of C-MEA-CCS (CoNiS//X37-50//CoNiS) prepared by different hot-pressing pressure (0, 0.2, 0.5 and 1.0 MPa) in 5 wt.% KOH at 60 °C. (e) The cell voltage of C-MEA-CCS (CoNiS//X37-50//CoNiS) with different hot-pressing pressure at various current density (250, 500 and 1000 mA cm⁻²). (f) Nyquist plot of MEAs with different hot-pressing pressure at 1.4 V_{cell} in 5 wt.% KOH at 60 °C.

Fig. R2 | Morphologies characterization of C-MEA-CCS (CoNiS//X37-50//CoNiS). SEM images of MEAs prepared by hot-pressing of (a) 0 MPa, (b) 0.2 MPa, (c) 0.5 MPa and (d) 1.0 MPa. (e) SEM images of the interface between the catalyst layer and membrane (position 1 marked in (d)). (f) SEM images of the catalyst layer (position 2 marked in (d)). (g) Corresponding EDX mapping of MEA prepared by hot-pressing of 1.0 MPa.

Fig. R3 | Performance of conventional MEAs (C-MEAs) and all-in-one MEA-S in 5 wt% and 0.5 wt% KOH. C-MEA-CCM (CoNiS//X37-50//CoNiS) assembled with commercial Sustainion X37-50 membrane and powdery CoNiS as cathode and anode; C-MEA-CCM (Pt/C//X37-50//IrO₂) assembled with Sustainion X37-50 membrane, powdery Pt/C as cathode and powdery IrO₂ as anode; C-MEA-CCS (CoNiS//X37-50//CoNiS) assembled with commercial Sustainion X37-50 membrane and self-supported CoNiS as cathode and anode. (a) Polarization curves of different MEAs in 5 wt.% KOH at 60 °C. (b) Nyquist plot of different MEAs at 0.5 A cm⁻² in 5 wt.% KOH at 60 °C. (c) Bar diagram of overvoltage current density (200, 500 and 1000 mA cm⁻²) of different MEAs divided by mass transfer (η_{mass}), kinetics (η_{kin}), and ohmic (η_{ohm}) in 0.5 wt.% KOH at 60 °C. (d) Polarization curves of different MEAs in 0.5 wt.% KOH at 60 °C. (e) Nyquist plot of different MEAs at 0.5 A cm⁻² in 0.5 wt.% KOH at 60 °C. (f) Bar diagram of overvoltage current density (200, 500 and 1000 mA cm⁻²) of different MEAs divided by mass transfer (η_{mass}), kinetics (η_{kin}), and ohmic (η_{ohm}) in 0.5 wt.% KOH at 60 °C. (g) Electrolysis stability for the conventional MEAs and all-in-one MEA-S in 0.5 wt.% KOH at 60 °C.

constant current density of 1000 mA cm^{-2} at $60 \text{ }^\circ\text{C}$.

We have made the following revisions in the main text:

(1) On page 2 of the revised manuscript:

In contrast, the hot-pressing process can be utilized to assemble CCS on membranes to achieve good CL/membrane interfacial contact in MEAs-CCS (**Supplementary Fig. 1 and 2**).

(2) We have also added this Fig. R1-3 as new Supplementary Fig. 1-2 in revised Supplementary Materials.

(3) We have also added Supplementary Note 1 in the revised Supplementary Materials:

Supplementary Note 1

To further investigate the impact of the interfacial resistance on the performance of conventional MEA-CCS (C-MEA-CCS) using Zirfon membrane and self-supported electrodes, we utilized the hot-pressing method to regulate the binding force between self-supported electrodes and membranes. As shown in Supplementary Fig. 2a-b, the introduction of hot pressing will increase the electrolysis performance of MEAs, especially at high current densities. Furthermore, to characterize the ohmic resistances in MEAs, EIS was performed at 1.4 V (Supplementary Fig. 2c). The intercept in the high-frequency region represents the ohmic resistance, which consists of the ionic and electronic resistances. Improving the hot-pressing pressure reduced the ohmic resistance of cell.

We also investigated the impact of the interfacial resistance on the performance of C-MEA-CCS using Sustainion X37-50 membranes and self-supported electrodes. Supplementary Fig. 1 exhibited SEM images of the AEM/electrode interface after hot pressing under different pressures. It can be observed that the interface distance decreased as improving the hot-pressing pressure from 0 to 1.0 MPa . When 30 wt\% KOH solution was used as the electrolyte, MEAs-CCS with different hot-pressing pressures showed discrepant electrolysis performance (Supplementary Fig. 2d). Typically, the MEA with 1.0 MPa showed cell voltage of 1.61 , 1.76 , and 2.03 V at different current densities of 500 ,

1000, and 2000 mA cm⁻², which were lower than that (1.67, 1.84, and 2.16 V) of the MEA with 0.1 MPa (Supplementary Fig. 2e). Furthermore, to characterize the ohmic resistances in MEAs, EIS was performed at 1.4 V (Supplementary Fig. 2f). Improving the hot-pressing pressure reduced the ohmic resistance of cell. This result indicates that reducing the interfacial resistance between the electrode and the membrane can obviously improve the electrolysis performance. When 10 wt% KOH solution was used as the electrolyte, the electrolysis performance difference of MEAs-CCS with different hot-pressing pressures was significant (Supplementary Fig. 2g). Typically, the MEA with 1.0 MPa showed cell voltage of 1.54, 1.68, and 1.92 V at different current densities of 250, 500, and 1000 mA cm⁻², which were significantly lower than that (1.65, 1.87, and 2.3 V) of the MEA with 0.1 MPa (Supplementary Fig. 2h). Improving the hot-pressing pressure significantly reduced the ohmic resistance of cells, especially at low-concentration KOH solution (Supplementary Fig. 2i).

Comment 3:

Also, it seems that the author should be clear whether the application of this development MEA is AEMWE or development for AEW. The MEA manufacturing methods and development directions in the two fields are different. Reviewers are confused by the presented data by the authors.

Response to the comments 3:

We appreciate the reviewer's valuable comments. Sorry for the confusion. We agree with the review's comments that the MEA manufacturing methods and development in AEMWE and alkaline liquid water electrolysis (ALWE) are different. As shown in Table R1, to date, self-supported electrodes and membranes have been widely used for ALWE. By contrast, the MEA in AEMWE is fabricated by sandwiching a membrane between the porous cathode and anode. Catalysts are applied to the MEA by the catalyst-coated membrane (CCM) and catalyst-coated substrate (CCS) methods. The key difference between ALWE and AEMWE is that the electrode separator in the former technology has no intrinsic ionic conductivity, its conductivity being provided by KOH filling the pores of a porous

separator in ALWE, while in AEMWE, the polymeric membrane is non-porous and possesses intrinsic anionic conductivity (*ChemSusChem*. 2022,15, e2022000). However, most studies on AEMWE are currently resorting to diluting KOH electrolyte support for reaching high performance.

As shown in Table R2, alkaline water electrolysis using novel all-in-one MEA has key advantages over conventional ALWE: (i) Remarkable electrolysis performance realized with low concentrations of electrolytes due to low ohmic resistance; (ii) With an *in-situ* pore-filled composite membrane, the hydrogen permeability is significantly reduced, thus allowing pressure operation. Moreover, alkaline water electrolysis using all-in-one MEA has key advantages over AEMWE: (i) Compared to conventional MEA-CCM, 3D-ordered CLs can be formed on the membrane in all-in-one MEA, significantly improving the catalyst utilization and accelerating gas/liquid transfer; (ii) The good alkaline resistance of PP and transition metal sulfide endows the PP composite membranes with highly alkaline stability, even at 30 wt% KOH; (iii) The integrated catalyst layer (CL) /membrane interfaces reduced the ohmic resistance of a single cell, especially at low concentrated KOH electrolyte. Moreover, the integrated CL/membrane interfaces avoided the formation of K_2CO_3 precipitate at the CL/membrane interfaces.

The present data in the manuscript has been added and corrected as follows:

- (1) Fig. R4 shows the area resistance of different membranes in the low-concentrated KOH were measured. Interestingly, all-in-one MEA-S exhibits area resistance of 0.19 and 0.30 $\Omega \text{ cm}^2$ in 5 and 0.5 wt% KOH, respectively, which is lower than that of Sustainion X37-50 (0.33 and 0.34 $\Omega \text{ cm}^2$), *m*-PBI (0.82 and 1.41 $\Omega \text{ cm}^2$) and Zirfon Thin (1.00 and 1.50 $\Omega \text{ cm}^2$).
- (2) The electrolysis performance of all-in-one MEA-S and C-MEAs-CCS with different commercial membranes (Zirfon Thin, *m*-PBI, and Sustainion X37-50) under different concentrated KOH solutions was compared. As shown in Fig. R5j, the all-in-one MEA-S exhibits a higher current density of 1.1 A cm^{-2} in 0.5 wt% KOH, in comparison with that of conventional MEA-CCS with Zirfon Thin (0.20 A cm^{-2}), *m*-PBI (0.21 A cm^{-2}) and X37-50 (0.45 A cm^{-2}) at 1.8 V. The total overvoltage is separated into ohmic overvoltage (η_{Ohm}), kinetic overvoltage (η_{kin}) and mass-transfer overvoltage (η_{mass}) (Fig. R5k). The ratios of η_{Ohm} of all-in-one MEA-S and conventional MEA-CCS with Zirfon Thin, *m*-PBI, and X37-50 were 19.0%, 36.2%, 35.6%, and 30.2% at 0.2 mA cm^{-2} .

², respectively. The difference between the η_{ohm} values of all-in-one MEA-S (23.6%) and conventional MEA-CCS with X37-50 (40.2%) was greater at a higher current density of 1 A cm^{-2} . Electrochemical impedance spectroscopy (EIS) was performed at a current density of 0.5 A cm^{-2} to analyze the differences in the performances of different MEAs (Fig. R51). The all-in-one MEA-S exhibited lower cell ohmic resistance (R_{ohm}), in comparison with conventional MEAs with Zirfon Thin and *m*-PBI membranes, which stemmed from the low area resistance of the composite membranes. In addition, the all-in-one MEA-S showed a lower R_{ohm} of $0.31 \text{ } \Omega \text{ cm}^2$ than that of conventional MEA with X37-50 ($0.42 \text{ } \Omega \text{ cm}^2$), which may be attributed to the integrated catalyst layer/membrane interface in all-in-one MEA-S.

- (3) The electrolysis performance of all-in-one MEA-S was compared with that of conventional MEAs (C-MEAs) prepared by the CCS or CCM method in 5 and 0.5 wt% KOH (Fig. R6).

Table R1. Comparison of conventional MEAs and all-in-one MEA for alkaline water electrolysis

	MEA-CCS		MEA-CCM	All-in-one MEA
Electrode	Powdery catalysts on the substrate	Self-supported electrode	Powdery catalysts on the membrane	Self-supported catalyst on membrane
Catalyst layer (CL)	Dense and inordered CL	3D-ordered CL	Dense and inordered CL	3D-ordered CL
Membrane	AEM	AEM	Diaphragm	AEM
Electrolyte	< 5 wt% KOH	< 5 wt% KOH	30 wt% KOH	< 5 wt% KOH
Electrolyzer	AEMWE	AEMWE	ALWE	AEMWE
References	Appl. Catal. B 2022,306,121127; Adv. Energy Mater ,2020,10,2002285	Nat Commun 2022,13,605	Chem. Eng. J. 2022,428,131149	Energy Environ. Sci. , 2021,14, 6338-6348
				This work

Table R2. Advantages and disadvantages of alkaline water electrolysis using conventional MEAs and all-in-one MEA

ALWE using MEA-CCS	AEMWE using MEA-CCS	AEMWE using MEA-CCM	Alkaline water electrolysis using all-in-one MEA
Commercial Zirfon Thin	Sustainion X37-50	Sustainion X37-50	PP/CoNiS composite membrane
Self-supported electrodes	Self-supported electrodes	Powdery CLs	3D-ordered CLs
High chemical stability of separator	Membrane degradation	Membrane degradation	High chemical stability of composite membrane
OH ⁻ transport by pore of separator	OH ⁻ transport by tethered cationic groups	OH ⁻ transport by tethered cationic groups	OH ⁻ transport by pore of the composite membrane
High gas permeation	Low gas permeation	Low gas permeation	Low gas permeation
Corrosive liquid electrolyte (30 wt.% KOH)	Pure water or low-concentrated electrolyte (1 M KOH)	Pure water or low-concentrated electrolyte (1 M KOH)	Low-concentrated electrolytes (0.5~30 wt% KOH)
High interfacial resistance, especially at low-concentrated KOH	High interfacial resistance, especially at low-concentrated KOH	Low interfacial resistance	Low interfacial resistance
3D-ordered and porous CLs	3D-ordered and porous CLs	Inordered and dense CLs	3D-ordered and porous CLs

Fig. R4 | Area resistance of the Zirfon Thin (200 μm), *m*-PBI (20 μm), Sustainion X37-50 (50 μm), and all-in-one MEA-S (~20 μm) as a function of KOH solution concentration.

Fig. R5 | Performance of conventional MEA-CCS (C-MEA-CCS) and all-in-one MEA-S. C-MEA-CCS (CoNiS//Zirfon Thin//CoNiS) uses Zirfon Thin and self-supported CoNiS as membrane and electrodes. C-MEA-CCS (CoNiS//*m*-PBI//CoNiS) uses *m*-PBI and self-supported CoNiS as membranes and electrodes. C-MEA-CCS (CoNiS//X37-50//CoNiS) uses Sustainion X37-50 and self-supported CoNiS as membrane and electrodes. Polarization curves of different MEAs in (a) 30 wt% KOH, (d) 10 wt% KOH, (g) 5 wt% KOH and (j) 0.5 wt% KOH at 60 °C. The cell voltage of different MEAs in (b) 30 wt% KOH, (e) 10 wt% KOH, and (h)

5 wt% KOH at various current densities. Nyquist plot of different MEAs at 500 mA cm⁻² in (c) 30 wt% KOH, (f) 10 wt% KOH, (i) 5 wt% KOH and (i) 0.5 wt.% KOH at 60 °C. (k) Bar diagram of overvoltage current density (200, 500, and 1000 mA cm⁻²) of different MEAs divided by mass transfer (η_{mass}), kinetics (η_{kin}), and ohmic (η_{ohm}) in 0.5 wt.% KOH at 60 °C.

Fig. R6 | Performance of conventional MEAs (C-MEAs) and all-in-one MEA-S in 5 wt% and 0.5 wt% KOH. C-MEA-CCM (CoNiS//X37-50//CoNiS) assembled with commercial Sustainion X37-50 membrane and powdery CoNiS as cathode and anode; C-MEA-CCM (Pt/C//X37-50//IrO₂) assembled with Sustainion X37-50 membrane, powdery Pt/C as cathode and powdery IrO₂ as anode; C-MEA-CCS (CoNiS//X37-50//CoNiS) assembled with commercial Sustainion X37-50 membrane and self-supported CoNiS as cathode and anode. (a) Polarization curves of different MEAs in 5 wt.% KOH at 60 °C. (b) Nyquist plot of different MEAs at 0.5 A cm⁻² in 5 wt.% KOH at 60 °C. (c) Bar diagram of overvoltage current density (200, 500, and 1000 mA

cm⁻²) of different MEAs divided by mass transfer (η_{mass}), kinetics (η_{kin}), and ohmic (η_{ohm}) in 0.5 wt.% KOH at 60 °C. (d) Polarization curves of different MEAs in 0.5 wt.% KOH at 60 °C. (e) Nyquist plot of different MEAs at 0.5 A cm⁻² in 0.5 wt.% KOH at 60 °C. (f) Bar diagram of overvoltage current density (200, 500, and 1000 mA cm⁻²) of different MEAs divided by mass transfer (η_{mass}), kinetics (η_{kin}), and ohmic (η_{ohm}) in 0.5 wt.% KOH at 60 °C. (g) Electrolysis stability for the conventional MEAs and all-in-one MEA-S in 0.5 wt.% KOH at a constant current density of 1000 mA cm⁻² at 60 °C.

We have made the following revisions in the main text:

(1) On page 10 of the revised manuscript:

Fig. 3f shows the area resistance of commercial membranes and all-in-one MEA-S as a function of KOH solution concentration. All-in-one MEA-S exhibited lower area resistance in KOH solutions of different concentrations compared with those of Zirfon Thin, *m*-PBI and Sustainion X37-50 membranes. The area resistance of commercial Zirfon membrane and *m*-PBI membrane sharply increased from 0.30 and 0.27 Ω cm² in 30 wt% KOH solution to 1.50 and 1.37 Ω cm² in 0.5 wt% KOH solution. In contrast, all-in-one MEA-S showed a loose upward trend in area resistance with decreasing electrolyte concentration. All-in-one MEA-S showed the lowest area resistances of 0.30 and 0.05 Ω cm² in 0.5 and 30 wt% KOH solutions, respectively, which were even lower than those of Sustainion X37-50 membrane (0.32 and 0.28 Ω cm²).

(2) On page 13 of the revised manuscript:

Fig. 4b compares the electrolysis performance of all-in-one MEA-S and alkaline liquid water electrolysis (ALWE) using C-MEA-CCS in 30 wt% KOH. The current density of all-in-one MEA-S was 2.5 A cm⁻² at 1.8 V, representing a 2.5-fold increase compared with C-MEA-CCS using Zirfon Thin and *m*-PBI membranes. The EIS spectrum (**Fig. 4d**) was used to investigate the contributions of ohmic and charge-transfer resistances. A perfect semicircle was obtained from all-in-one MEAs, whereas the presence of a semicircle at low frequency originating from mass-transfer resistance was observed for C-MEA-CCM³⁴. The impedance at high frequency, intercepting the *x*-axis, represented the cell ohmic resistance (R_{ohm})³⁵. The estimated initial area-specific resistances of all-in-one MEA-S and C-MEA-CCS comprising Zirfon Thin and *m*-PBI membranes were equal to 0.09, 0.30 and 0.35 Ω cm² in 30 wt% KOH, respectively, which agreed with the *ex-situ* area resistance data. The cell ohmic resistance results also suggested that the electrolyzer was not a short circuit. All-in-one MEA-S exhibited a smaller charge-transfer resistance (R_{ct}) compared with that of C-MEA-CCS, manifesting better electrochemical reaction kinetics. When low-concentration KOH electrolyte (0.5-10 wt%)

was used, all-in-one MEA-S showed notably superior performance compared with C-MEA-CCS comprising Zirfon Thin and *m*-PBI membranes (**Supplementary Fig. 38**). The electrolysis performance of all-in-one MEA-S and AEM water electrolysis (AEMWE) with C-MEAs using a Sustainion X37-50 membrane was also compared (**Fig. 4c**). When 5 wt% KOH electrolyte was used, all-in-one MEA-S showed superior performance (1.45 A cm⁻² at 1.8 V) compared with C-MEA-CCM (CoNiS//X37-50//CoNiS), C-MEA-CCM (Pt/C//X37-50//IrO₂), and C-MEA-CCS (CoNiS//X37-50//CoNiS) (0.52, 0.50, and 0.41 A cm⁻² at 1.8 V, respectively). Furthermore, the EIS measurements of MEAs were compared (**Fig. 4e**). All-in-one MEA-S exhibited lower R_{ohm} and R_{ct} values than those of C-MEAs in 5 wt% KOH. The low cell resistances of all-in-one MEA-S stemmed from the integrated CL/membrane interfaces and well-ordered OH⁻ transfer channel in membrane. These results indicated that alkaline water electrolysis using all-in-one MEA was more efficient than ALWE and AEMWE using C-MEAs. Notably, the performance of all-in-one MEA-S fed with 0.5 wt% KOH remained higher than that of C-MEAs with the Sustainion X37-50 membrane (**Supplementary Fig. 39**).

(4) We have added this Fig. R5 as new Supplementary Fig. 38 in the Supplementary Materials.

(5) We have replotted Fig. 4 in the revised manuscript:

Fig. 4 | Electrolysis performance of all-in-one MEAs. (a) Schematic diagram of advanced alkaline water electrolysis using the all-in-one MEA. Carbon paper and Ni foam were used as the cathode and anode liquid/gas diffusion layers, respectively. (b) Polarization curves of C-MEA-CCS (CoNiS//Zirfon Thin//CoNiS), C-MEA-CCS (CoNiS//m-PBI//CoNiS), and all-in-one MEA-S in 30 wt% KOH at 60 °C. C-MEA-CCS (CoNiS//Zirfon Thin//CoNiS) and C-MEA-CCS (CoNiS//m-PBI//CoNiS) were prepared using self-supported CoNiS electrodes. (c) Polarization curves of C-MEA-CCM (CoNiS//X37-50//CoNiS), C-MEA-CCM (Pt/C//X37-50//IrO₂), C-MEA-CCS (CoNiS//X37-50//CoNiS), and all-in-one MEA-S in 5 wt% KOH at 60 °C. C-MEA-CCS (CoNiS//X37-50//CoNiS) was prepared using self-supported CoNiS electrodes. (d) Nyquist impedance plots of different MEAs in 30 wt% KOH at 60 °C. (e) Nyquist impedance plots of different MEAs in 5 wt% KOH at 60 °C. (f) Comparison of advanced alkaline water electrolysis using all-in-one MEA-S with reported ALWE, PGM-free AEMWE, and PEMWE. (g) Durability cell voltage-time plots for C-MEA-CCM and all-in-one MEA-S at a constant current density of 1000 mA cm⁻² at 60 °C.

Comment 4:

For example, m-PBI, FAA-3, and zirfone were compared, but the explanation of how MEA was fabrication (What kind of ionomer was used for each) and compared is insufficient. What is the cell temperature? What is the thickness of the manufactured all in one MEA? Have you considered the performance change according to the thickness of m-PBI, FAA, and zirfon used? m-PBI and FAA used are high-density polymer electrolyte membranes and zirfon is a diaphragm. Is an integral MEA a porous diaphragm? Or is it a kind of polymer electrolyte membrane that gas does not pass through? In addition, Why didn't you compare the sustain membrane of Dioxide Materials, which is known to have the best performance in AEMWE?

Response to the comments 4:

Comment 4.1: For example, m-PBI, FAA-3, and zirfone were compared, but the explanation of how MEA was fabrication (What kind of ionomer was used for each) and compared is insufficient.

Response: We thank the reviewer for the valuable suggestion. As suggested by the reviewer, we added detail information in the manuscript and supplementary materials, including the fabrication of MEAs

and electrolysis measurements. C-MEAs were prepared using different fabrication methods, including CCS and CCM. For C-MEA-CCM preparation, commercial 40% Pt/C (Johnson Matthey) and IrO₂ (Alfa Aesar, MA, USA) catalysts were employed as HER and OER electrocatalysts, respectively. The cathode and anode inks were prepared according to recent literature. For anode ink preparation, IrO₂ powder (5 mg) was mixed with DI water (50 mg), isopropyl alcohol (100 mg), and quaternary ammonia poly(N-methyl-piperidine-co-p-terphenyl) (QAPPT) ionomer (5.5 mg; 10 wt%). The anode ionomer ratio was 10 wt%. For cathode ink preparation, Pt/C powder (5 mg) was mixed with DI water (25 mg), isopropyl alcohol (50 mg) and QAPPT ionomer (12.5 mg; 10 wt%). The cathode ionomer ratio was 20 wt%. The disperse inks were sprayed onto both sides of an AEM (Sustainion X37-50). The IrO₂ and Pt/C loadings were 2.0 and 0.5 mg cm⁻², respectively. The cathode carbon paper and anode Ni foam were sandwiched with the catalyst coated membrane and pressed to form a MEA at 70 °C for 3 min at 0.1 MPa. C-MEAs-CCS with powder catalysts were prepared using the same catalyst ink and loading steps as performed for C-MEA-CCM. The disperse ink was sprayed on Ni foam and carbon paper, forming anode and cathode, respectively. The as-prepared electrodes were pressed with a membrane (including PP, PES, and Sustainion X37-50 membranes) at 70 °C for 3 min at 0.1 MPa. To prepare C-MEA-CCS with self-supported electrodes, the synthesis process of the self-supported CoNiS electrode was the same as that used for all-in-one MEA-S, but replacing the porous PP membrane with Ni foam or carbon paper. The as-prepared self-supported electrodes were pressed with a membrane (such as PES, Zirfon Thin, m-PBI, and Sustainion X37-50) at 70 °C for 3 min at 0.1 MPa. The loadings of cathode and anode CLs using powder CoNiS catalysts on the membrane or liquid/gas diffusion layer were both 2.5 mg cm⁻².

As shown in Fig. R7 and R8, the electrolysis performance of C-MEA-CCS using Zirfon Thin, *m*-PBI and Sustainion X37-50, and all-in-one MEA was further compared. To verify the KOH concentration effect, polarization curves and corresponding EIS of different MEAs are measured as a function of KOH concentration. The lower area resistance of the Sustainable X37-50 membrane is likely contributing to its higher cell performance compared with Zirfon Thin and *m*-PBI membranes. Specifically, the all-in-one MEA-S has the highest cell performance under different KOH concentrations (30 wt% to 0.5 wt%) due to the lowest area resistance and integrated catalyst layer/membrane interface.

In addition, the cell performance of C-MEA-CCS with Zirfon Thin and *m*-PBI membranes is highly pH dependent, which could be attributed to the following reasons: 1) The OER and HER activity for the alkaline water electrolysis significantly increased as the KOH concentration increased (*Nat Energy* 2020, 5, 378–385); 2) The ability of the porous Zirfon Thin membrane to provide ionic conductivity by allowing the passage of the electrolyte. Thus, the area resistance of the Zirfon Thin membrane is highly pH dependent; 3) *m*-PBI membrane forms a homogeneous ternary electrolyte system of polymer/water/KOH, which utilize the uptake and presence of an aqueous alkaline electrolyte to achieve ionic conductivity (*Energy Environ. Sci.*, 2019,12, 3313-3318). Thus, the area resistance of the *m*-PBI membrane is also highly pH dependent. Notably, the all-in-one MEA-S still shows a high current density of 1.40 A cm⁻² at a cell voltage of 1.80 V at 60 °C in 5 wt% KOH (Fig. R7g-h). However, the MEAs with Zirfon Thin, *m*-PBI, and X37-50 membranes displays a low current density of 0.35, 0.40, and 0.53 A cm⁻² at 1.80 V. EIS measurements were performed to evaluate the cell resistance (R_{ohm}) of MEAs under different KOH concentration (Fig. R8). The R_{ohm} of all-in-one MEA-S increases from 0.08 Ω cm² in 30 wt% KOH to 0.32 Ω cm² in 0.5 wt% KOH, which is lower than those (0.34 and 0.46 Ω cm² in 30 and 0.5 wt% KOH, respectively) of C-MEA-CCS with X37-50 membrane. By contrast, the R_{ohm} of C-MEA-CCS with Zirfon Thin and *m*-PBI membranes significantly increases as decreasing KOH solution concentration. Therefore, the results suggest that integrated catalyst layer/membrane interfaces and well-ordered OH⁻ transfer channels in all-in-one MEA can decrease cell resistance and improve cell performance.

Fig. R7 | Performance of conventional MEA-CCS (C-MEA-CCS) and all-in-one MEA-S. C-MEA-CCS (CoNiS//Zirfon Thin//CoNiS) uses Zirfon Thin and self-supported CoNiS as membrane and electrodes. C-MEA-CCS (CoNiS//*m*-PBI//CoNiS) uses *m*-PBI and self-supported CoNiS as membranes and electrodes. C-MEA-CCS (CoNiS//X37-50//CoNiS) uses Sustainion X37-50 and self-supported CoNiS as membrane and electrodes. Polarization curves of different MEAs in (a) 30 wt% KOH, (d) 10 wt% KOH, (g) 5 wt% KOH and (j) 0.5 wt% KOH at 60 °C. The cell voltage of different MEAs in (b) 30 wt% KOH, (e) 10 wt% KOH, and (h)

5 wt% KOH at various current densities. Nyquist plot of different MEAs at 500 mA cm⁻² in (c) 30 wt% KOH, (f) 10 wt% KOH, (i) 5 wt% KOH and (i) 0.5 wt% KOH at 60 °C. (k) Bar diagram of overvoltage current density (200, 500, and 1000 mA cm⁻²) of different MEAs divided by mass transfer (η_{mass}), kinetics (η_{kin}), and ohmic (η_{ohm}) in 0.5 wt.% KOH at 60 °C.

Fig. R8 | Cell resistances (R_{ohm}) of all-in-one MEA-S and C-MEA-CCS with different membranes as a function of KOH solution concentration. The R_{ohm} values were obtained from high-frequency resistance in EIS plots at 60 °C.

Comment 4.2: What is the thickness of the manufactured all-in-one MEA? Have you considered the performance change according to the thickness of m-PBI, FAA, and zirfon used?

Response: As shown in Fig. R9a, the thickness of the pristine PP membrane is 20 μm . After the oriented intergrowth of the catalyst layer, the thickness of the as-prepared all-in-one MEA-S is $\sim 26 \mu\text{m}$ (Fig. R9b). As shown in Fig. R10, the area resistance of membrane will decrease with the thickness.

Commercial Zirfon membranes with two different thickness are available, including Zirfon PERL (500 μm) and Zirfon Thin (200 μm) (Fig. R10a). As shown in Fig. R10b, the thickness of commercial FAA-3-50 and FAA-3-20 membranes is 50 and 20 μm , respectively. The thickness of commercial Sustainion X37-50 is 50 μm . Moreover, the area resistance of the Sustainion X37-50 membrane is lower than that of the FAA-3-20 membrane. In addition, the area resistance of *m*-PBI membranes with different thicknesses of 20, 50, and 100 μm was evaluated (Fig. R10c). Therefore, in this manuscript, the commercial Zirfon Thin, Sustainion X37-50, and *m*-PBI (20 μm) membranes are chosen to prepare C-MEAs.

Fig. R9 | Cross-sectional SEM images. (a) pristine PP porous separator and (b) all-in-one MEA-S.

Fig. R10 | Area resistance of different membranes. (a) Area resistance of the Zirfon PERL (500 μm) and Zirfon Thin (200 μm) as a function of KOH solution concentration. (b) Area resistance of the FAA-3-50 (50 μm) and FAA-3-20 (20 μm) as a function of KOH solution concentration. (c) Area resistance of *m*-PBI membranes with different thicknesses of 20, 50, and 100 μm as a function of KOH solution concentration.

Comment 4.3: *m*-PBI and FAA used are high-density polymer electrolyte membranes and zirfon is

a diaphragm. Is an integral MEA a porous diaphragm? Or is it a kind of polymer electrolyte membrane that gas does not pass through?

Response: We thank the reviewer for the insightful and constructive comments. The integral MEA is a composite membrane with small-sized nanopores. The pristine PP membrane exhibits typical elliptical pore structures. After pore-filling treatment, the pores were densely filled with CoNiS nanosheets. To further understand the membrane structure in all-in-one MEA-S, we newly conducted mercury intrusion porosimetry (MIP) and capillary flow porometry (CFP) to analyze the pore structures of the membrane in all-in-one MEA-S. The mercury-pore size distributions are shown in Fig. R11. PP and Zirfon Thin membranes showed a bimodal distribution in sizes of approximately 30-60 nm and 100-1000 nm. By contrast, all-in-one MEA-S showed a weak peak intensity in the whole region. These results indicate that the pore-filling method significantly decreases the pore size and distribution of membrane. MIP is not suitable for investigating gas permeation through porous membranes because the result is also affected by the presence of blind pores (*J. Power Sources*. 2022,524,231059). In contrast, CFP selectively measures only the constricted through pores of the membrane. In this study, the CFP experiment is performed in wet test. The air pressure is gradually applied to the wet-treated membrane. When a certain pressure is reached, the fluid filling the largest pore is pushed out first, and the flow occurs. The differential pressures and flow rates of PP separator, Zirfon Thin, and all-in-one MEA-S are shown in Fig. R12a. For PP separator, the pressure at bubble point is 0.82 bar. The mean pore diameter of PP separator is 0.30 μm (Fig. R12b). The Zirfon Thin membrane exhibits higher pressure (1.31 bar) than that of PP separator at bubble point. All-in-one MEA-S shows a high pressure of 11.1 bar at bubble point due to the low average pore diameter of ~ 32 nm (Fig. R12d). From the above results, the all-in-one MEA-S can significantly improved gas barrier properties.

Fig. R11 | The pore size distribution of different membranes and all-in-one MEA-S. The curves of incremental intrusion curves for pristine PP separator, Zirfon Thin and all-in-one MEA-S as a function of pore size diameter by mercury porosimeter.

Fig. R12 | The bubble point and pore size distribution of different membranes and all-in-one MEA-S. (a) Differential pressures and flow rates of PP separator, Zirfon Thin, and all-in-one MEA-S. (b) The pore diameter distribution of (b) PP separator, (c) Zirfon Thin and (d) all-in-one MEA-S, measured using a capillary flow porometer.

Comment 4.4: In addition, Why didn't you compare the sustain membrane of Dioxide Materials, which is known to have the best performance in AEMWE?

Response:

Firstly, we should explain that the commercial FAA-3-20 membrane is compared due to frequently-used anion exchange membrane. In this work, we considered that different types of membranes or diaphragms were compared with our all-in-one MEAs due to different OH⁻ transport behavior. For example:

As a typical ion-solvating membrane, *m*-PBI membrane utilizes the uptake of KOH electrolyte

to achieve hydroxide conductivity.

As a typical anion exchange membrane, FAA-3-20 membrane conducts OH⁻ by tethered cationic groups via a combination of Grotthus and vehicular mechanisms.

As a commercial Zirfon diaphragm, the alkaline electrolyte is transported through the pores of diaphragms.

More importantly, as suggested by reviewer, the FAA-3-20 membrane is replaced by the Sustainion X37-50 membrane (Dioxide Materials) for comparison. We have newly made the following experiments:

(1) The basic property of Sustainion X37-50 membrane, including area resistance, alkaline stability and mechanical strength.

As shown in Fig. R13, the all-in-one MEA-S has higher tensile strength and elongation at break than the Zirfon Thin, *m*-PBI and Sustainion X37-50 membranes. Fig. R14 shows area resistance of different membranes as a function of KOH solution concentration. The all-in-one MEA-S exhibits lower area resistance in different concentrated KOH solution than those of Zirfon Thin, *m*-PBI and Sustainion X37-50 membranes. Interestingly, all-in-one MEA-S still exhibits a lower area resistance of 0.30 Ω cm² compared to that (0.33 Ω cm²) of Sustainion X37-50 in a low concentrated KOH solution (0.5 wt%). To further obtain a powerful insight into OH⁻ transport behavior in the composite membrane, the ion transference number through the membrane was analyzed and calculated (Fig. R15). Compared with the ion transference numbers of commercial Zirfon Thin, *m*-PBI and Sustainion X37-50 membranes, the composite membrane in all-in-one MEA-S demonstrated higher hydroxide ion conductivity. The alkaline stability of the composite membrane was studied in 30 wt% KOH at 80 °C (Fig. R16), in which the composite membrane showed a negligible change in the area resistance after 1000 h alkaline treatment. As expected above, the excellent chemical stability of composite membrane in all-in-one MEA-S resulted from the ether-free polymer backbone and chemically stable metal sulfide. However, the commercial Zirfon Thin, *m*-PBI and Sustainion membranes exhibited 80% and 30% area resistance increase rate after 1000 h of the durability test. In addition, the change in the mechanical properties of the membranes after alkaline stability testing was probed. The commercial Zirfon Thin, *m*-PBI and Sustainion membranes lost 80% (from 25 to 5 MPa) and 47% (from 36 to 20 MPa) tensile strength after 1000 h of the durability test. By contrast, the composite membrane in all-

in-one MEA-S exhibits a negligible change in the mechanical properties. Finally, as shown in Fig. R17, compared with representative commercial membranes, the composite membrane in all-in-one MEA-S exhibited overwhelming advantages for alkaline water electrolysis from the aspects of the area resistance, swelling, stress strength, alkaline stability and gas barrier properties.

(2) As responded to the comments 2 and comments 3, the electrolysis performance of Sustainion X37-50 membrane has been added.

Fig. R13 | The mechanical strength of membranes and all-in-one MEA-S. Stress-strain curves of the PP separator, Zirfon Thin, *m*-PBI, Sustainion X37-50 membranes and all-in-one MEA-S.

Fig. R14 | Area resistance of the Zirfon Thin, *m*-PBI, Sustainion X37-50 and all-in-one MEA-S as a function of KOH solution concentration. The area resistance tests were conducted at 60 °C.

Fig. R15 | The ionic transport properties of different membranes and all-in-one MEA-S. The current-voltage profile of Zirfon Thin, *m*-PBI, Sustainion X37-50 membranes and all-in-one MEA-S measured in a KOH concentration gradient of 1-3 mol L⁻¹.

Fig. R16 | Alkaline stability of different membranes and all-in-one MEA-S. Area resistance remained and mechanical strength of different membranes under different treatment time of 200, 500 and 1000 h in 30 wt.% KOH at 80 °C

Fig. R17 | Comprehensive comparison of the properties of different membranes and all-in-one MEA-S. The area resistance, swelling ratio, stress strength, bubble point and degradation of commercial Zirfon Thin,

m-PBI, Sustainion X37-50 membranes and all-in-one MEA-S are compared.

We have made the following revisions in the main text:

(1) We have corrected the data using commercial Sustainion X37-50 membrane, thus new Fig. 3 is provided in the revised manuscript:

Fig. 3 | Basic properties from membrane characterization. Cross-sectional SEM images of (a) pristine PP porous separator and (b) all-in-one MEA-S, and corresponding elemental mapping. Inset: High-resolution SEM images. The 3D-reconstruction image projection of (c) pristine PP porous separator and (d) all-in-one MEA-S. (e) Contact angle and bubble point pressure of pristine PP porous separator, Zirfon Thin and all-in-one MEA-S. (f) Area resistance of commercial Zirfon Thin, *m*-PBI, and Sustainion X37-50 membranes, and all-in-one MEA-S. (g) Current-voltage profiles of Zirfon Thin, *m*-PBI, Sustainion X37-50 membranes and all-in-one MEA-S measured using a KOH concentration gradient of 1-3 mol L⁻¹. (h) Area resistance and mechanical strength of different membranes at different treatment times of 200, 500 and 1000 h in 30 wt% KOH at 80

°C. (i) Comprehensive comparison of the properties of different membranes.

(2) We have also added this Fig. R11-12 as new Supplementary Fig. 34-35 in revised Supplementary Materials.

Comment 5:

Caption of Figure and scheme with SI have to be modified according to the manuscript. For example, in Figure 1d there is no lamellar structure. This can be seen in 1e. Figure S11 also does not match the text. Even Figure S30 and S31 have the same caption.

Response to the comments 5:

We appreciate the reviewer's valuable comments. Sorry for the confusion. The caption of Figure and scheme with SI has been modified according to the manuscript.

We have made the following revisions in the main text:

(1) On page 5 of the revised manuscript:

The uniform nanosheets structure of CoNiS was confirmed by transmission electron microscopy (TEM; Fig. 2d).

Comment 6:

Experimental Section also have to be modified. More detailed explanation needed. For example, The sample names for the case of using pp porous membrane and the case of using PE-reinforced PTFE are used interchangeably. Also what is MEA-R, CoNiS NW etc? In addition, there is no detailed description of the MEA manufacturing method in these cases, and the description of each analysis and experimental method (Ex. Interfacial bonding strength, porosity of all in one MEA) is insufficient. With the current experimental method written by the author, it is difficult to reproduce.

Response to the comments 6:

We appreciate the reviewer's valuable comments. Sorry for the confusion. Experimental Section has been thoroughly supplemented and corrected.

We have made the following revisions in the maintext:

(1) On page 23 of the Experimental Section in the revised manuscript:

Preparation of all-in-one MEA-Z

The synthetic route to all-in-one MEA-Z followed the same preparation steps used to obtain all-in-one MEA-S, but replacing the PP porous membrane with a commercial Zirfon Thin membrane.

Preparation of all-in-one MEA-R

The all-in-one MEA-R was prepared using a PE-reinforced PTFE porous membrane with a pore size of 0.1 μm , high porosity of $> 90\%$, and thickness of 100 μm . The synthetic route to all-in-one MEA-R followed the same preparation steps used to obtain all-in-one MEA-S, but replacing the PP porous membrane with a PE-reinforced PTFE porous membrane.

Preparation of all-in-one MEA-NS

The synthetic route to all-in-one MEA-NS followed the same preparation steps used to obtain all-in-one MEA-S, but replacing $\text{Co}(\text{NO}_3)_2 \cdot 6\text{H}_2\text{O}$ and $\text{Ni}(\text{NO}_3)_2 \cdot 6\text{H}_2\text{O}$ with $\text{CoCl}_2 \cdot 6\text{H}_2\text{O}$ and $\text{NiCl}_2 \cdot 6\text{H}_2\text{O}$.

Preparation of all-in-one MEA (CoNiS@FeNi LDH)

For the preparation of all-in-one MEA (CoNiS@FeNi LDH), the precursor solution was prepared by mixing $\text{Ni}(\text{NO}_3)_2 \cdot 6\text{H}_2\text{O}$ (1.6 mmol), $\text{FeSO}_4 \cdot 7\text{H}_2\text{O}$ (0.4 mmol), urea (8 mmol), and H_2O (60 mL). Then, the precursor solution containing as-prepared all-in-one MEA-S was treated by a hydrothermal method at 100 $^\circ\text{C}$ for 3 h.

(2) On page 24 of the Experimental Section in the revised manuscript:

Conventional MEAs (C-MEAs) preparation

C-MEAs were prepared using different fabrication methods, including CCS and CCM. For C-MEA-CCM

preparation, commercial 40% Pt/C (Johnson Matthey) and IrO₂ (Alfa Aesar, MA, USA) catalysts were employed as HER and OER electrocatalysts, respectively. The cathode and anode inks were prepared according to recent literature¹⁴. For anode ink preparation, IrO₂ powder (5 mg) was mixed with DI water (50 mg), isopropyl alcohol (100 mg), and quaternary ammonia poly(N-methyl-piperidine-co-p-terphenyl) (QAPPT) ionomer (5.5 mg; 10 wt%). The anode ionomer ratio was 10 wt%. For cathode ink preparation, Pt/C powder (5 mg) was mixed with DI water (25 mg), isopropyl alcohol (50 mg) and QAPPT ionomer (12.5 mg; 10 wt%). The cathode ionomer ratio was 20 wt%. The disperse inks were sprayed onto both sides of an AEM (Sustainion X37-50). The IrO₂ and Pt/C loadings were 2.0 and 0.5 mg cm⁻², respectively. The cathode carbon paper and anode Ni foam were sandwiched with the catalyst coated membrane and pressed to form a MEA at 70 °C for 3 min at 0.1 MPa. C-MEAs-CCS with powder catalysts were prepared using the same catalyst ink and loading steps as performed for C-MEA-CCM. The disperse ink was sprayed on Ni foam and carbon paper, forming anode and cathode, respectively. The as-prepared electrodes were pressed with a membrane (including PP, PES, and Sustainion X37-50 membranes) at 70 °C for 3 min at 0.1 MPa. To prepare C-MEA-CCS with self-supported electrodes, the synthesis process of the self-supported CoNiS electrode was the same as that used for all-in-one MEA-S, but replacing the porous PP membrane with Ni foam or carbon paper. The as-prepared self-supported electrodes were pressed with a membrane (such as PES, Zirfon Thin, *m*-PBI, and Sustainion X37-50) at 70 °C for 3 min at 0.1 MPa. The loadings of cathode and anode CLs using powder CoNiS catalysts on the membrane or liquid/gas diffusion layer were both 2.5 mg cm⁻².

(3) On page 24 of the Experimental Section in the revised manuscript:

Physical characterization

The micromorphology and nanostructure of all-in-one MEAs were analyzed by field-emission SEM (FE-SEM, Merlin) and TEM (JEM2010, Japan). To observe the CL/membrane interface, a TESCAN S9000 SEM instrument equipped with focused ion beam (FIB) accessories was used. The FIB was used for surface etching and further in-depth elemental analysis. The surface structure of CLs was analyzed by means of a white light interferometry profiling system (Bruker Contour GT K0 G, Germany). AFM (MFP-3D-SA, USA) was utilized to determine the CoNiS nanosheet thickness. Crystallographic information for CoNi LDH and CoNiS was obtained by XRD using Cu K α irradiation ($\lambda = 1.5406 \text{ \AA}$). Compositional information was obtained by XPS

(Ulvac-Phi, Japan) using a Thermo Scientific K-Alpha spectrometer. X-ray micro-computed tomography (Xradia 620 Versa) was used to obtain 3D structures of samples. Nitrogen adsorption-desorption isotherms were tested on an Autosorb-IQ2-MP-C system at 77 K. The specific surface area was evaluated following the multipoint Brunauer-Emmett-Teller (BET) method. The average pore diameter and porosity of the separators or membranes were determined using MIP (AutoPore Iv 9510) and CFP (PMI, USA). The interfacial bonding strength of MEAs was measured using an electronic universal material testing machine (WDW-100). The contact angles were measured by Contact Angle Tester (Data physics Co. Ltd., GER OCA-20). More physical characterization of membranes can be found in the Supplementary Materials.

Comment 7:

CoNi-LDH material is a very popular material in supercapacitors under alkaline conditions. It is also used as an amphoteric catalyst in alkaline water electrolysis. Also, CoNi-LDH material has been grown on various supports (ZIF, Porous polymer substrate, etc) and used as capacitors or catalysts. Therefore, it seems that the novelty of this study needs more explanation. Is this the first time this material has been applied to alkaline water electrolysis?

Response to the comments 7:

We really appreciate the reviewer's efforts on our work and thank the reviewer for providing insightful comments to further improve the quality of our manuscript. Regarding these comments, we would like to clarify below the reviewer's concerns, which we hope to make this manuscript powerful to the high standard of Nature Communications. We acknowledge that the used catalytic materials have been widely studied. However, in this paper, much attention has been paid on the development of novel and 3D-ordered membrane electrode assemblies (MEAs). Critically different from the previous works that catalysts materials grown on porous polymer substrate (For example: Adv. Energy Mater.2020, 10, 1901892). The inorganic materials grown on various supports were only used as individual catalysts. By contrast, in this manuscript, the polymer substrate also acts as a membrane for OH⁻ conduction and gas separation in MEA. To the best of our knowledge, in this paper for the first

time, CoNi LDH or CoNiS has been grown in porous membrane, forming all-in-one MEAs applied in alkaline water electrolysis. Our work had made very important achievements as follows:

(1) Actually, developing MEAs with high performance and durability is key to promoting the wide applications of alkaline water electrolysis. To date, the development focus in numerous studies, so far, has been either on improving the conductivity of membranes or on investigating highly active catalysts. However, the integration of the electrocatalyst in the catalytic layer and in the MEA is critical for performance and durability improvements, but also critical to allow switching from dilute KOH or K_2CO_3 . In this paper for the first time, we have directly introduced 3D-ordered electrocatalysts (including CoNi LDH, CoNiS and CoNiS@FeNi LDH) on membrane, forming all-in-one MEA with integrated catalyst layer/membrane interface for alkaline water electrolysis.

(2) Very important, Using platinum-group-metal (PGM) free catalysts, the present all-in-one MEAs achieved high current density of 5.0 and 2.2 $A\ cm^{-2}$ at 2.0 V with 30 wt% and 5 wt% KOH, respectively, which surpasses that of state-of-the-art ALWE and AEMWEs. PGM-free all-in-one MEAs operated stably under a 1000 $mA\ cm^{-2}$ current density at 60 °C for more than 1000 h.

(3) To note that: the CoNiS was introduced into porous membrane to form all-in-one MEA, of which CoNiS can be replaced by other novel electrocatalysts with high-performance and stability, further improving the electrolysis performance.

Therefore, we believe that this paper could open a new pathway for next-generation MEAs for advanced alkaline water electrolysis and other energy conversion applications (such as CO_2 electroreduction).

Comment 8:

After sulfuration (CoNi-LDH to CoNiS), There is no evidence of more roughness.(Figure 1c)

Response to the comments 8:

We appreciate the reviewer's valuable comments. The related sentences "the surface of the nanosheets become very rough" have been deleted.

We have made the following revisions in the main text:

(1) On page 4 of the revised manuscript:

As shown in Fig. 2c, the main structure of the nanosheet arrays after sulfurization was almost unchanged.

Comment 9:

The reviewer is really curious, in the case of AEW and AEMWE, the diaphragm or the electrolyte membrane plays a role in blocking the electrical conductivity between the two electrodes, respectively. However, in the all-in one MEA manufactured by the author, there is no diaphragm or zero gap to block electrical conductivity. The current all-in-one MEA is considered a short circuit system. It was understood that CoNi-LDH or CoNiS material acts as an amphoteric catalyst and shows high activity for HER and OER, showing high performance in water electrolysis. However, how can the electrical conductivity between the two electrodes be blocked in the current system? Where is the electrolyte membrane (or diaphragm) in the system now, and where are the electrodes? Where does each electrode reaction take place? A detailed explanation is needed for this. In Scheme 1b and c, the CoNi-LDH or CoNiS material filled with the porous PP membrane, as if it were a different material from the electrode, was displayed in different colors. However, it is correct to express it in the same color as the electrode.

Response to the comments 9:

We thank the reviewer for their careful consideration of our work. Let us first explain the electron transport in the all-in-one MEA-S. As shown in Fig. R18, the electron can be transported into the catalyst layer through gas/liquid diffusion layer. However, the electron can not be further transported through composite membrane because of the discontinuous CoNiS nanosheets and the insulation role of PP support. The diameter of CoNiS nanosheets in composite membrane is $\sim 3 \mu\text{m}$. Thus, we consider that the electron can not efficient transport through discontinuous conductive channel in a composite membrane with a thickness of $\sim 20 \mu\text{m}$. In addition, inorganic Mg-Al layered double hydroxide (LDH) as membrane for water electrolysis has been reported (Nano Energy (2015) 11, 110–118; <https://www.sciencedirect.com/science/article/pii/S2211285514202199>). To further address this concern, we newly conducted the experiments. The all-in-one MEA is insulating for electron transport in the dry state. Additionally, the Faradaic efficiency is measured to verify if all electron went into water

electrolysis (Fig. R19). The volumes of H₂ and O₂ produced by all-in-one MEA-S at different current density of 1000 and 4000 mA cm⁻² were measured, giving a nearly 100% Faraday efficiency. As shown in Fig. R20 and Fig. R21, as suggested, the colors have been corrected.

Fig. R18 | Schematic of the behavior of electron and ion transport in the all-in-one MEA.

Fig. R19 | Faradaic efficiency. Volume of H₂ and O₂ generation by the advanced alkaline water electrolysis using the all-in-one MEA-S at different current density of 1000 and 4000 mA cm⁻² under atmospheric pressure.

Fig. R20 | Schematic illustration of concept and preparation process of all-in-one MEA. (a) Conventional MEA (C-MEA) and (b) all-in-one MEA based on 3D-ordered transport highways. (c) The synthesis procedure of all-in-one MEA.

Fig. R21 | Schematic diagram of the advanced alkaline water electrolysis using the all-in-one MEA. The carbon paper and Ni foam are used as the cathode and anode liquid/gas diffusion layer, respectively.

We have made the following revisions in the main text:

(1) On page 16 of the revised manuscript:

To verify if all electron were utilized in water electrolysis, the volumes of H₂ and O₂ produced by all-in-one MEA-S were measured at different current densities of 1000 and 4000 mA cm⁻² (**Fig. 5f**), showing almost 100% Faraday efficiency.

Comment 10:

If the manufactured three-dimensionally aligned CoNi-LDH or CoNiS electrode shows good performance, how about forming this electrode separately and evaluating the MEA performance compared to the all-in-one MEA?

Response to the comments 10:

We appreciate the reviewer's valuable comemnts. To form the electrode separately and evaluating the MEA performance, we prepare the CoNi LDH or CoNiS nanoarrays on porous Ni foam, in which the fabrication method is identical except for the substrate. More importantly, the catalyst loading on Ni foam maintains the same with that on Ni foil. In addition, we newly provided the HER and OER performance of CoNi-LDH or CoNiS on porous Ni foam as below.

Fig. R22 | Electrochemical properties for OER and HER. (a) OER polarization curves of CoNiS/Ni foam, CoNiS/Ni foil, CoNi LDH/Ni foam, CoNi/Ni foil, Ni foam and Ni foil in 1.0 M KOH at a scan rate of 5 mV s⁻¹. (b) The corresponding Tafel plots. (c) HER polarization curves of CoNiS/Ni foam, CoNiS/Ni foil, CoNi LDH/Ni foam, CoNi/Ni foil, Ni foam and Ni foil in 1.0 M KOH at a scan rate of 5 mV s⁻¹. (d) The corresponding Tafel plots.

We have made the following revisions in the main text:

(1) On page 9 of the revised manuscript:

In addition, when Ni foil was replaced with porous Ni foam, the obtained electrochemical performance of electrodes was almost same (**Supplementary Fig. 25**).

(2) We have added this Fig. R22 as new Supplementary Fig. 25 in the revised Supplementary Materials.

Comment 11:

The evidence is needed for the levelized cost of hydrogen (LCOH) and the calculated energy efficiency.

Response to the comments 11:

We thank the reviewer for their suggestion. The evidence for the LCOH and calculated energy efficiency has been added as follow.

1. Definition of energy efficiency:

The energy efficiency of an electrolysis cell is defined as the net energy present in the hydrogen produced by the cell divided by the net energy consumed by the cell to produce it, expressed as a percentage. The net energy present in hydrogen is its *higher heating value* (HHV), which is 39.4 kWh kg⁻¹ of hydrogen. An electrolysis cell operating at a cell potential equal to the thermoneutral voltage of water electrolysis (1.48 V at room temperature, 1.47 V at 60 °C) displays 100% energy efficiency. Therefore, $\eta = 1.47 \text{ (V)}/E_{\text{cell}}$, where η and E_{cell} are the energy efficiency and voltage of cell, respectively.

2. Levelized cost of hydrogen (LCOH) analysis:

The LCOH was used as an index to quantify the cost of hydrogen production from alkaline water electrolysis. The LCOH (\$ kg⁻¹) can be expressed as follows:

$$LCOH = \frac{(CAPEX \times CRF) + C_{O\&M}}{M_{H_2}}$$

Where CAPEX is the total capital expenditure of the electrolyzer (\$), CRF is the capital recovery factor, C_{O&M} is the annual operation and maintenance costs (\$) and M_{H₂} is the total hydrogen produced by the electrolyzer in one year (kg).

CRF stands for the capital recovery factor, which converts the capital cost into a series of equivalent annual payments over the system lifetime N considering an interest rate i.

$$CRF = \frac{i \times (1 + i)^N}{(1 + i)^N - 1}$$

The capital costs of the electrolyzer CAPEX can be calculated with following equation:

$$CAPEX = P_{el} \times I_{el}$$

Where P_{el} is the rated power of the electrolyzer (kW) and I_{el} is the specific investment cost of the electrolyzer (\$ kW⁻¹).

The annual operation and maintenance costs $C_{O\&M}$ (\$) include the costs of electricity, electrolyte, nonfuel variable operation and maintenance.

$$C_{O\&M} = (\tau \times P_{el} \times \mu_{el} \times c_e) + (\gamma \times M_{H_2} \times c_w) + (CAPEX \times \varphi)$$

Where τ is the total number of hours in the year (h), P_{el} is the rated power of the electrolyzer (kW), μ_{el} is the utilization rate of the electrolyzer expressed as a fraction of 1, c_e is the price of electricity (\$ kWh⁻¹), γ is the water required to produce each kg of hydrogen (L kg⁻¹), M_{H_2} is the hydrogen produced by the installation in one year (kg), and c_w is the price of electrolyte (\$ kg⁻¹). Maintenance costs are assumed to be constant throughout the system's lifetime and are estimated as a fraction (φ) of the electrolyzer capital cost.

The annual production of hydrogen using alkaline water electrolysis can be computed using following equation:

$$M_{H_2} = \frac{\tau \times P_{el} \times \mu_{el}}{E_{el}}$$

In the equation above, E_{el} stands for the power consumption of the electrolyzer (kWh kg⁻¹).

3. Common assumptions:

Advanced alkaline water electrolysis stack system is constructed by stacking 200 single cells (200-cell) (Table R3). The stack provided a current density of 1000 mA cm⁻² at approximately 392, 356, 330 and 314 V_{stack} for conventional MEA-CCM (Pt/C//X37-50//IrO₂), conventional MEA-CCS (CoNiS//Zirfon Thin//CoNiS), all-in-one MEA and all-in-one MEA-S, respectively.

Conventional MEA-CCM (Pt/C//X37-50//IrO₂): electricity cost = \$25 MWh⁻¹, the specific investment cost of the electrolyzer I_{el} = \$400 kW⁻¹, electrolyzer lifetime = 10 years, interest rate (i) = 6.5%, operations and maintenance (O&M) cost = 4% of electrolyzer CAPEX p.a., cost of electrolyte (5 wt% KOH) = \$0.002 kg⁻¹.

Conventional MEA-CCS (CoNiS//Zirfon Thin//CoNiS): electricity cost = \$25 MWh⁻¹, the specific investment cost of the electrolyzer I_{el} = \$300 kW⁻¹, electrolyzer lifetime = 20 years, interest

rate (i) = 6.5%, operations and maintenance (O&M) cost = 4% of electrolyzer CAPEX p.a., cost of electrolyte (30 wt% KOH) = \$0.01 kg⁻¹.

All-in-one MEA-S and all-in-one MEA (CoNiS@FeNi LDH): electricity cost = \$25 MWh⁻¹, the specific investment cost of the electrolyzer $I_{el} = \$200 \text{ kW}^{-1}$, electrolyzer lifetime = 20 years, interest rate (i) = 6.5%, operations and maintenance (O&M) cost = 4% of electrolyzer CAPEX p.a., cost of electrolyte (30 wt% KOH) = \$0.01 kg⁻¹, cost of electrolyte (5 wt% KOH) = \$0.002 kg⁻¹.

Fig. R23 shows the comparison of LCOH of a electrolyzer stack based on conventional MEA-CCM (Pt/C//X37-50//IrO₂), conventional MEA-CCS (CoNiS//Zirfon Thin//CoNiS), all-in-one MEA-S and all-in-one MEA (CoNiS@FeNi LDH). The LCOH of all-in-one MEA-S and all-in-one MEA (CoNiS@FeNi LDH) is lower than that of conventional MEA-CCM (Pt/C//X37-50//IrO₂) and conventional MEA-CCS (CoNiS//Zirfon Thin//CoNiS) in various aspects of electricity, maintain and capital expenditure cost.

Table R3. Comparison of operating parameters of stacks of 200 of the individual cells at 60 °C

Cell stack type	Fixed current density / A cm ⁻²	Number of cells	Current / A	Cell stack voltage / V	Total power / kW	Electrolyte / KOH wt%
Conventional MEA-CCM (Pt/C//X37-50//IrO ₂)	1.0	200	200	392	78.4	5
Conventional MEA-CCS (CoNiS//Zirfon Thin//CoNiS)	1.0	200	200	356	71.2	30
All-in-one MEA-S	1.0	200	200	330	66.0	5
All-in-one MEA (CoNiS@FeNi LDH)	1.0	200	200	314	62.8	30

Fig. R23 | The LCOH of advanced alkaline water electrolysis. The LCOH of C-MEA-CCM (Pt/C//X37-50//IrO₂), C-MEA-CCS (CoNiS//Zirfon Thin//CoNiS), all-in-one MEA-S and all-in-one MEA (CoNiS@FeNi LDH) was compared.

We have made the following revisions in the main text:

(3) On page 21 of the revised manuscript:

A key strategy to reduce the levelized cost of hydrogen (LCOH) involves improving the energy efficiency of alkaline water electrolyzers. **Fig. 6c** shows a comparison of the LCOH of electrolyzer stacks based on C-MEA-CCS (CoNiS//Zirfon Thin//CoNiS), C-MEA-CCM (Pt/C//X37-50//IrO₂), and all-in-one MEA-S. The LCOH of all-in-one MEA-S was lower than that of C-MEAs in categories including electricity, maintenance, and capital expenditure costs. Therefore, the all-in-one MEAs show potential to improve the competitiveness of green hydrogen with fossil fuels.

(4) On page 5-8 of the revised Supplementary Materials: The evidence for the LCOH and calculated energy efficiency has been added.

(5) We have added this Fig. R21 as new Fig. 6c in the revised manuscript.

Comment 12:

Manuscript needs substantial English polishing and typos revision.

Response to the comments 12:

We thank the reviewer for their suggestion. As suggested, we have carefully checked and polished our manuscript in the revised version by a native English speaker.

To Reviewer 2:

Referee 2:

The authors of ‘Biomimetic approach to the oriented intergrowth of catalyst layer in membrane electrode assembly for alkaline water electrolyzer’ presented a study in which the bifunctionality of their intergrown 3D catalyst layers with advanced performance for alkaline water electrolysis was showcased. The authors made a step-by-step representation of their catalyst (CoNi LDH CoNiS) and growth onto PP porous membrane later replaced by PE reinforced PTFE membrane and use of CoNiS@FeNi LDH as catalyst. The paper carries value as the fusing of electrode and membrane should bring about a decrease in interfacial resistances and improved performances beyond that of CCM design. It would be of interest to see the performance of such an all-in-one MEA design also reported under pressure operation in future works for AEMWE. There are comments and the appeal to authors to have the manuscript submitted for language editing. There are typos, untypical word choices and incorrect use of prepositions (as example) present throughout the manuscript and SI. I support the publication of manuscript after major revision.

Response to the comments:

We are grateful for the time and effort. Reviwer 2 has reviewed our manuscript, which are constructive and valuable to improve our manuscript.

Comment 1:

21: add the KOH concentration the measurement was performed at here. It is relevant when interpreting AWE or AEMWE performances. Cell potential recorded at 1 A/cm² in 30 wt% KOH.

Response to the comments 1:

The authors thank the reviewer for this helpful comment. The KOH concentration has been added.

We have made the following revisions in the main text:

(1) On page 1 of the revised manuscript:

Owing to this design, a high current density of 1000 mA cm⁻² was obtained at 1.57 V in 30 wt% KOH, equating to 94% energy efficiency..

Comment 2:

29-30: 'A porous membrane permeable to hydroxide ions prevents gas permeability'. – Disagree with the use of 'prevents' in this context – should rather be limits or decreases gas permeability and hence mixing of O₂ and H₂ product gases. H₂ crossover still occurs during operation.

Response to the comments 2:

The authors agree with the reviewer for this comment. The use of 'prevents' has been corrected.

We have made the following revisions in the main text:

(1) On page 2 of the revised manuscript:

A porous membrane permeable to hydroxide ions limits gas permeability.

Comment 3:

'fall off' – rather to refer to detachment of catalyst here.

Response to the comments 3:

We thank the reviewer for this comment. We have corrected this mistake.

We have made the following revisions in the main text:

(1) On page 2 of the revised manuscript:

Finally, some catalyst could be easily peeled off from the electrode at a high current density, leading to low electrolysis stability.

Comment 4:

reference 12 refers mainly to PEMs. Consider to add references for AEM studies conducted on CCM methods.

Response to the comments 4:

The authors thank the reviewer for this comment. As suggested by reviewer, the reference 12 has been replaced by references for AEM studies conducted on CCM methods. For example, *ACS Applied Energy Materials* 4, 1053-1058 (2020); *Energy & Environmental Science* 14, 6338-6348 (2021)

We have made the following revisions in the maintext:

(1) On page 3 of the revised manuscript:

However, these *in-situ* CL deposition techniques cannot be used to form MEAs-CCM on polymer membranes owing to the nonconducting, swelling, and thermolabile properties of polymer membranes^{12, 13, 14}.

13. Soni R, Miyanishi S, Kuroki H, Yamaguchi T. Pure Water Solid Alkaline Water Electrolyzer Using Fully Aromatic and High-Molecular-Weight Poly(fluorene-alt-tetrafluorophenylene)-trimethyl Ammonium Anion Exchange Membranes and Ionomers. *ACS Applied Energy Materials* 4, 1053-1058 (2020).

14. Chen N, Paek SY, Lee JY, Park JH, Lee SY, Lee YM. High-performance anion exchange membrane water electrolyzers with a current density of 7.68 A cm⁻² and a durability of 1000 hours. *Energy & Environmental Science* 14, 6338-6348 (2021).

Comment 5:

Scheme 1: The arrows for OH⁻ directed from anode in (a) and (b) are misleading. OH⁻ species will be conducted from cathode to the anode.

Response to the comments 5:

The authors thank the reviewer for this comment. The arrows for OH⁻ conduction have been corrected.

Fig. R24 | Schematic illustration of concept and preparation process of all-in-one MEA. (a) Conventional MEA (C-MEA) and **(b)** all-in-one MEA based on 3D-ordered transport highways. **(c)** Synthetic procedure for all-in-one MEA-S.

We have made the following revisions in the maintext:

(1) We have added this Fig. R24 as new Fig. 1 in the revised manuscript.

Comment 6:

65: add KOH concentration operated in here as well.

Response to the comments 6:

The authors thank the reviewer for this comment. The KOH concentration has been added.

We have made the following revisions in the maintext:

(1) On page 4 of the revised manuscript:

Therefore, advanced alkaline water electrolyzers employing this all-in-one MEA show a high current density of 1000 mA cm⁻² at 1.57 V **in 30 wt% KOH**. Importantly, the all-in-one MEA operated stably at a high current density of 1000 mA cm⁻² at 60 °C for more than 1000 h **in 30 wt% KOH**.

Comment 7:

102: Add reference in text to the methods section where more information is shared on the conventional CCM. (refer to 425) does not seem to be the optimal catalyst ink formulation for well performing MEA. Which ionomer was used? No water added to the solvent mixture? Also include thickness of membrane here.

Response to the comments 7:

The authors thank the reviewer for this comment. As suggested, the conventional MEA-CCM (C-MEA-CCM) is renewedly prepared by the optimal catalyst ink formulation. The C-MEA-CCM was prepared and tested again following the reported literature (*Energy Environ. Sci.*, 2021,14, 6338-6348). For C-MEA-CCM preparation, commercial 40% Pt/C (Johnson Matthey) and IrO₂ (Alfa Aesar, MA, USA) catalysts were employed as HER and OER electrocatalysts, respectively. The cathode and anode inks were prepared according to recent literature¹⁴. For anode ink preparation, IrO₂ powder (5 mg) was mixed with DI water (50 mg), isopropyl alcohol (100 mg), and quaternary ammonia poly(N-methyl-piperidine-co-p-terphenyl) (QAPPT) ionomer (5.5 mg; 10 wt%). The anode ionomer ratio was 10 wt%. For cathode ink preparation, Pt/C powder (5 mg) was mixed with DI water (25 mg), isopropyl alcohol (50 mg) and QAPPT ionomer (12.5 mg; 10 wt%). The cathode ionomer ratio was 20 wt%. The disperse inks were sprayed onto both sides of an AEM (Sustainion X37-50). The IrO₂

and Pt/C loadings were 2.0 and 0.5 mg cm⁻², respectively.

We have made the following revisions in the maintext:

(2) On page 23-24 of the revised manuscript:

C-MEAs were prepared using different fabrication methods, including CCS and CCM. For C-MEA-CCM preparation, commercial 40% Pt/C (Johnson Matthey) and IrO₂ (Alfa Aesar, MA, USA) catalysts were employed as HER and OER electrocatalysts, respectively. The cathode and anode inks were prepared according to recent literature¹⁴. For anode ink preparation, IrO₂ powder (5 mg) was mixed with DI water (50 mg), isopropyl alcohol (100 mg), and quaternary ammonia poly(N-methyl-piperidine-co-p-terphenyl) (QAPPT) ionomer (5.5 mg; 10 wt%). The anode ionomer ratio was 10 wt%. For cathode ink preparation, Pt/C powder (5 mg) was mixed with DI water (25 mg), isopropyl alcohol (50 mg) and QAPPT ionomer (12.5 mg; 10 wt%). The cathode ionomer ratio was 20 wt%. The disperse inks were sprayed onto both sides of an AEM (Sustainion X37-50). The IrO₂ and Pt/C loadings were 2.0 and 0.5 mg cm⁻², respectively.

Comment 8:

108: ref 20 & 21. Worthy references, but it should be kept in mind the OH⁻ transport mechanism for AEM electrolyzers (liquid&gas phase) does pose different challenges to that of PEM fuel cells (gas phase).

Response to the comments 8:

The authors thank the reviewer for this comment. We agree with the review's comments.

Comment 9:

142: There is mention of thickness of the catalyst growth, but is there any indication (estimated) of the loading for anode and cathode for the all-in-one membranes? Or which is later then electrochemically compared with other OER and HER catalysts (Fig 1 I & j) in terms of LSV an CA measurements?

Response to the comments 9:

We thank the referee for the valuable suggestion. The loading amount of CoNiS in all-in-one MEA

was about 5.3 mg cm⁻², which consisted of anode CL, cathode CL and pore-filling CoNiS. As shown in Fig. R25, the overpotential at j_{10} of CoNiS for OER and HER was compared with the reported bifunctional electrocatalysts.

Fig. R25 | The bifunctional activity comparison of CoNiS and reported bifunctional electrocatalysts.

Comparison of HER and OER overpotentials at 10 mA cm⁻² of this work with the reported bifunctional electrocatalysts, including Ni₁₁(HPO₃)₈(OH)₆ (*Energy Environ. Sci.*2018,11,1287-1298), Co-P-B (*Appl. Catal. B.*2019,259, 118051), CoFeO@BP (*Angew. Chem. Int. Ed.*2020,59, 21106-21113), Cr-doped FeNi-P/NCN (*Adv. Mater.*2019,31,e1900178), Cu-NiS₂ (*Small.*2020,16,e1905885), Fe-Ni₅P₄/NiFeOH (*Appl. Catal. B.* 2021,291,119987), Cu₃P-Cu₂O/NPC (*Chem. Eng. J.*2022,427, 130946), CoMnP/Ni₂P (*J. Mater. Chem. A.*2021,9,22129-22139), LSC/K-MoSe₂ (*Nat. Commun.*2021,12,4606), FeP/Ni₂P (*Nat Commun.*2018,9,2551), Cu₃N (*ACS Energy Lett.*2019,4,747-754), CoTe₂@NCNTFs (*J. Mater. Chem. A.* 2018,6,3684-3691), R-NCO (*J. Am. Chem. Soc.*2018,140,13644-13653), NiSe/NF (*Angew. Chem. Int. Ed.*2015,54,9351-9355), NiFeOF (*ACS Catal.*2017,7,8406-8412), NiFe-NCs (*ACS Appl. Mater. Interfaces.*2017,9, 41906-41915), NiFe-LDH@CoS_x (*Chem. Eng. J.*2021,419,129512), FeCoNi-HNTAs (*Nat. Commun.*2018,9,2452) and Co/CNFs (1000) (*Adv. Mater.*2019,31,e1808043).

Comment 10:

199-201: It seems the use of increased and decreased area resistance (or 30 and 10 wt% KOH exchanged in the sentence) in the sentence should be swapped . Please read again.

Response to the comments 10:

The authors thank the reviewer for this comment. This mistake has been corrected as “The area resistance of commercial Zirfon membrane and *m*-PBI membrane sharply increased from 0.30 $\Omega \text{ cm}^2$ and 0.27 $\Omega \text{ cm}^2$ in the 30 wt% KOH solution to 1.50 $\Omega \text{ cm}^2$ and 1.37 $\Omega \text{ cm}^2$ in the 0.5 wt% KOH solution.”

We have made the following revisions in the maintext:

(1) On page 10 of the revised manuscript:

The area resistance of commercial Zirfon membrane and *m*-PBI membrane sharply increased from 0.30 and 0.27 $\Omega \text{ cm}^2$ in 30 wt% KOH solution to 1.50 and 1.37 $\Omega \text{ cm}^2$ in 0.5 wt% KOH solution.

Comment 11:

155: Was it tried to grow the 3D-ordered CLs onto porous diaphragm Zirfon besides from the AEM (FAA-3-20) and *m*-PBI mentioned in the study?

Response to the comments 11:

We appreciate the reviewer’s valuable comments. We have tried to grow the 3D-ordered CLs onto porous Zirfon Thin membrane, forming all-in-one MEA-Z. Fig. R26a-b shows the cross-sectional SEM images of the commercial Zirfon Thin membrane. The aggregated mixtures of ZrO_2 nanoparticles and polysulfone (PSU) species are randomly distributed in Zirfon. Micro-sized pores were formed between ZrO_2 and PSU. As shown in Fig. R26c, the surface of the Zirfon membrane exhibited a porous structure. CoNiS nanosheet arrays can be formed on both sides of a porous Zirfon membrane through the oriented CL intergrowth strategy (Fig. R26d-f). Moreover, the micro-sized pores were densely filled with CoNiS nanosheets. From the EDS mapping on the cross-section, the CoNiS nanosheets are uniformly distributed in the Zirfon composite membrane (Fig. R26g). The conventional MEA-CCS (C-MEA-CCS) using self-supported CoNiS electrodes and Zirfon Thin membrane is prepared for comparison. Fig. R27 shows polarization curves recorded between 1.2 and 2.3 V_{cell} at 60 °C in 30 wt% KOH solution. The performance of the all-in-one MEA-Z far exceeded that of C-MEA-CCS (CoNiS//Zirfon Thin//CoNiS), especially at high current density. The C-MEA-CCS (CoNiS//Zirfon Thin//CoNiS) reached a current density of 1.51 A cm^{-2} at the cell voltage of 2.0 V. In contrast, the all-in-one MEA-Z achieved a high current density of 3.35 A cm^{-2} at the same cell voltage. Therefore, the strategy that oriented intergrowth of catalyst layer is efficient for porous diaphragm

Zirfon Thin membrane.

Fig. R26 | Morphological structure characterization of all-in-one MEA prepared using commercial Zirfon Thin membrane. (a-b) Cross-sectional and (c) surface SEM image of porous Zirfon membrane (inset shows the optical image). (d-e) Cross-sectional and (f) surface SEM image of all-in-one MEA-S based on Zirfon membrane (inset shows the optical image). (g) Cross-sectional SEM image of all-in-one MEA-S based on Zirfon membrane prepared by focused ion beam (FIB) and corresponding EDX mapping.

Fig. R27 | Electrolysis performance of C-MEA-CCS (CoNiS//Zirfon Thin//CoNiS) and all-in-one MEA-Z. Performance was measured at 60 °C with 30 wt% KOH circulating in the anode and cathode.

We have made the following revisions in the maintext:

(1) On page 20 of the revised manuscript:

The CoNiS nanosheet arrays can also be formed on a porous Zirfon Thin membrane, forming all-in-one MEA-Z (Supplementary Fig. 48).

(2) We have added this Fig. R26 as new Supplementary Fig. 47 in the revised Supplementary Materials.

(3) We have added this Fig. R27 as new Fig. 6b in the revised manuscript.

Fig. 6 | Generality of oriented CLs intergrowth strategy . Polarization curves of all-in-one MEA-R, all-in-one MEA-Z, all-in-one MEA-NW and all-in-one MEA (CoNiS@FeNi LDH) in 30 wt.% KOH at 60 °C.

Comment 12:

176: There is mention of earth abundant cobalt, however the mining conditions and exploiting of the workers deserve consideration in this context while it is anticipated that the battery market will increase their cobalt demands in future, driving up the price and scarcity of cobalt.

Response to the comments 12:

We appreciate the reviewer’s valuable comemnts. We agree with your comments. This sentence has been corrected as “The outstanding activity and durability made CoNiS a promising bifunctional electrocatalyst for advanced alkaline water electrolysis”

We have made the following revisions in the maintext:

(1) On page 9 of the revised manuscript:

This outstanding activity and durability make CoNiS a promising bifunctional electrocatalyst for advanced alkaline water electrolysis.

Comment 13:

230: Here the mention of including Zirfon in membrane characterization. Could the authors add the porosity /pore size of the material in comparison to porous PP used?

Response to the comments 13:

We appreciate the reviewer's valuable comments. The porosity/pore size of Zirfon separator has been added. To further understand the membrane structure in all-in-one MEA-S, we newly conducted mercury intrusion porosimetry (MIP) and capillary flow porometry (CFP) to analyze the pore structures of the membrane in all-in-one MEA-S. The mercury-pore size distributions are shown in Fig. R28. PP and Zirfon Thin membranes showed a bimodal distribution in sizes of approximately 30-60 nm and 100-1000 nm. By contrast, all-in-one MEA-S showed a weak peak intensity in the whole region. These results indicate that the pore-filling method significantly decreases the pore size and distribution of membrane. The differential pressures and flow rates of PP separator, Zirfon Thin, and all-in-one MEA-S are shown in Fig. R29a. For PP separator, the pressure at bubble point is 0.82 bar. The mean pore diameter of PP separator is 0.30 μm (Fig. R29b). The Zirfon Thin membrane exhibits higher pressure (1.31 bar) than that of PP separator at bubble point. All-in-one MEA-S shows a high pressure of 11.1 bar at bubble point due to the low average pore diameter of ~ 32 nm (Fig. R29d). From the above results, the all-in-one MEA-S can significantly improved gas barrier properties. The pore size and porosity of different membranes were summarized in Table. R4.

Fig. R28 | The pore size distribution of different membranes and all-in-one MEA-S. The curves of incremental intrusion curves for PP separator, Zirfon Thin and all-in-one MEA-S as a function of pore size diameter by mercury porosimeter.

Fig. R29 | The bubble point and pore size distribution of different membranes and all-in-one MEA-S. (a) Differential pressures and flow rates of PP separator, Zirfon Thin, and all-in-one MEA-S. (b) The pore diameter distribution of (b) PP separator, (c) Zirfon Thin and (d) all-in-one MEA-S, measured using a capillary flow porometer.

Table R4. Summary for the porosity and pore size of different membranes

	Porosity / %	BET pore size / μm	MP pore size / μm	CFP pore size / μm
Porous PP	42	0.34	0.36	0.30
Zirfon Thin	52	0.42	0.40	0.45
All-in-one MEA-S	4	0.04	0.05	0.03

We have made the following revisions in the maintext:

(1) On page 9 of the revised manuscript:

Mercury intrusion porosimetry (MIP) and capillary flow porometry (CFP) were also used to analyze the pore

structures of all-in-one MEA-S. All-in-one MEA-S exhibited an average pore diameter of ~63 nm, which was significantly lower than those of PP (~0.30 μm) and commercial Zirfon Thin (~0.45 μm) membranes

(2) We have added this Fig. R28 and Fig. R29 as new Supplementary Fig. 34 and Fig. 35 in the revised Supplementary Materials.

(3) We have added this Table. R4 as new Supplementary Table. 1 in the revised Supplementary Materials.

Comment 14:

Fig 3 (e)/ line 270: Are the references for these SoA data reported on AEM and PEMWE results listed in the SI? It could be required to add the references for the presented studies in this graph. For example are the AEMWE data points reported for non-PGM electrode studies only, or is AEMWE performances for KOH electrolyte supported studies only reported? Recent works in pure water have also proved promising, such as <https://doi.org/10.1038/s41560-020-0577-x> and <https://doi.org/10.1021/acscatal.0c04200>.

Response to the comments 14:

We appreciate the reviewer's valuable comments. As suggested, the references have been added in the revised manuscript. The performance comparison has been improved. As shown in Fig. R30, the alkaline water electrolysis employing all-in-one MEA-S fed with 30 and 5 wt% KOH outperformed most of the previously reported ALWE and AEMWE without platinum group metal (PGM). More importantly, the electrolysis performance of all-in-one MEA-S fed 0.5 wt% (0.1 M) KOH is further evaluated. As shown in Fig. R31, the performance of all-in-one MEA-S is still higher than those of C-MEAs with Sustainion X37-50 membrane. Importantly, the performance of all-in-one MEA-S fed with 0.5 wt% KOH is comparable with that of AEMWE in 0.5 wt% KOH and pure water (Fig. R32).

Fig. R30 | The electrolysis performance comparison of all-in-one MEA-S and reported literatures. Comparison of cell performances of advanced alkaline water electrolysis (AWE) using all-in-one MEA-S in 5 wt% and 30 wt% KOH in this study and the literatures (ALWE, AEMWE using precious platinum group metal (PGM) free catalysts and PEMWE).

Table R5. Comparison of cell performances of advanced alkaline water electrolysis (AWE) using all-in-one MEA-S in 5 wt% and 30 wt% KOH in this study and the literatures (ALWE, AEMWE using precious platinum group metal (PGM) free catalysts and PEMWE).

	Anode	Cathode	Membrane	Electro-lyte	Current density / A cm ⁻²	Cell voltage / V	Ref.
	IrO ₂	Pt/C	NPBI	6 M KOH	1.50	1.92	1
ALWE	NiFe LDH	Raney Ni	Zirfon-type	30 wt% KOH	0.60	1.70	2
	Ni foam	Ni foam	PVA/ABPBI	15 wt% KOH	0.50	1.93	3
	NiFe LDH	Raney Ni	Zirfon-type	30 wt% KOH	1.40	2.10	4

	NiFe LDH	Raney Ni	cPVA/Zirfon	30 wt% KOH	1.00	2.05	5
	Ni	Ni	Porous PFSA	30 wt% KOH	0.20	2.12	6
	Ni/MoN/rNS	Ni/MoN/rNS	PPS	30 wt% KOH	0.40	1.85	7
	Ni ₂ P@FePO _x H _y	MoNi ₄ /MoO ₂	AEM	1 M KOH	0.50	1.75	8
	NiFe	Ni	FAA-3-50	1 M KOH	0.40	2.00	9
AEMWE	Ni felt	Ni felt	Aemion™	1 M KOH	0.30	2.10	10
using PGM free catalysts	NiFeCo LDH	NiFeCoP	X37-50 Grade T	1 M KOH	0.50	1.72	11
	NiFe ₂ O ₄	NiFeCO	TPNPiQA	1 M KOH	0.85	2.20	12
	NiFe	NiFe	PFTP-13	1 M KOH	0.92	1.80	13
	NiFe LDH	NiMo	PVBC-Mpy/PEK	1 M KOH	0.90	2.13	14
	NiCo ₂ O ₄	NiCo ₂ O ₄	PAni	1 M KOH	0.37	2.00	15
	Ir	Pt/C	Nafion 115	Pure water	1.95	1.95	16
	IrO ₂	Pt/C	Nafion 117	Pure water	1.50	1.80	17
PEMWE	IrO ₂	Pt/C	Multiblock PEM	Pure water	3.00	1.87	18
	IrO ₂	Pt/C	Nafion 117	Pure water	2.00	1.80	19
	IrO ₂	CuNiMo	Nafion 212	Pure water	1.45	1.90	20
	IrO ₂	Pt/C	Nafion 115	Pure water	3.80	2.20	21

References in Table R5:

1. *J. Membr. Sci.* **2022**, 643, 120042
2. *J. Membr. Sci.* **2020**, 616, 118541
3. *J. Membr. Sci.* **2017**, 535, 45–55
4. *Chem. Eng. J.* **2022**, 428, 131149
5. *J. Power Sources.* **2022**, 524, 231059
6. *J. Membr. Sci.* **2015**, 493, 589–598
7. *Adv. Sci.* **2022**, 9, 2105869

8. *Appl. Catal. B.* **2022**, 306, 121127
9. *Chem. Eng. J.* **2022**, 433, 133774
10. *J. Mater. Chem. A.* **2022**, 10, 16061–16070
11. *Appl. Catal. B.* **2021**, 294, 120246
12. *J. Mater. Chem. A.* **2022**, 10, 8401–8412
13. *Energy Environ. Sci.*, **2021**, 14, 6338–6348
14. *J. Mater. Chem. A.* **2019**, 7, 17914–17922
15. *J. Mater. Chem. A.* **2020**, 8, 17089–17097
16. *Energy Environ. Sci.*, **2022**, 15, 109–122
17. *ACS Appl. Mater. Interfaces.* **2022**, 14, 1, 2335–2342
18. *J. Membr. Sci.* **2021**, 638, 119694
19. *Adv. Sci.* **2021**, 8, 2102950
20. *J. Power Sources.* 2021, 506, 230200
21. *Adv. Energy Mater.* 2021, 11, 2100630

Fig. R31 | Performance of conventional MEAs (C-MEAs) and all-in-one MEA-S. C-MEA-CCM (CoNiS//X37-50//CoNiS) uses Sustainion X37-50 and powdery CoNiS catalysts as membrane and CLs. C-MEA-CCM (Pt/C//X37-50//IrO₂) uses Sustainion X37-50, powery Pt/C and IrO₂ catalysts as membrane, cathode and anode CLs, respectively. C-MEA-CCS (CoNiS//X37-50//CoNiS) uses Sustainion X37-50 and self-supported CoNiS as membrane and electrodes. (a) Polarization curves of different MEAs in 0.5 wt.% KOH at 60 °C. (b) Nyquist plot of different MEAs at 0.5 A cm⁻² in 0.5 wt.% KOH at 60 °C. (c) Bar diagram of overvoltage current density (200, 500 and 1000 mA cm⁻²) of different MEAs divided by mass transfer (η_{mass}), kinetics (η_{kin}) and ohmic (η_{ohm}) in 0.5 wt.% KOH at 60 °C.

Fig. R32 | The electrolysis performance comparison of all-in-one MEA-S and reported literatures. Comparison of cell performances of advanced alkaline water electrolysis (AWE) using all-in-one MEA-S in 0.5 wt% KOH in this study and the literatures (AEMWE fed with 0.1 M KOH and pure water).

Table R6. Comparison of cell performances of advanced alkaline water electrolysis (AWE) using all-in-one MEA-S in 0.5 wt% KOH in this study and the literatures (AEMWE fed with 0.1 M KOH and pure water).

	Anode	Cathode	Membrane	Electro-lyte	Current density / A cm ⁻²	Cell voltage / V	Ref.
AEMWE	a-NiFeOOH	MoNi ₄ /MoO ₂	FAA-3-50	Pure water	0.25	1.88	1
	NiFe	Ni	FAA-3-50	0.1 M KOH	0.31	2.00	2
	Ni-LCO/C	Cu _{0.5} Co _{2.5} O ₄	X37-50 Grade T	0.1 M KOH	0.78	1.90	3
	NiFe LDH	Pt/C	HWU-AEM	Pure water	0.25	1.90	4
	NiCoFeO _x	Pt black	AEM	Pure water	0.58	2.25	5
	NiFe-BTC-GNPs	MoNi ₄ /MoO ₂	FAA-3-PK-130	0.1 M KOH	1.10	1.85	6
	NiCoFe-NDA	Pt/C	X37-50 Grade T	0.1 M KOH	0.42	1.90	7
	IrO ₂	Pt/C	PiperION	Pure water	0.80	1.97	8

IrO ₂	Pt/C	QPC-TMA	Pure water	0.14	1.70	9
FeNiOOH-F	Pt/C	PAP-TP-85	Pure water	1.02	1.80	10
Ir-Ni/Mo ₅ N ₆	Ir-Ni/Mo ₅ N ₆	X37-50 Grade T	Pure water	0.11	1.6	11
Acta 3030	Acta 4030	A201	1 wt% K ₂ CO ₃	0.47	1.81	12
IrO ₂	Pt black	A201	Pure water	0.39	1.80	13
IrO ₂	Pt/C	PFOTFPh	Pure water	1.00	1.96	14

References in Table R6:

1. *J. Mater. Chem. A*, **2021**,9, 14043-14051
2. *Chem. Eng. J.* **2022**,433, 133774
3. *ACS Sustainable Chem. Eng.* **2021**, 9, 37, 12508–12513
4. *J. Mater. Chem. A*, **2021**,9, 23485-23496
5. *ACS Catal.* **2019**, 9, 1, 7–15
6. *Energy Environ. Sci.*, **2020**,13, 3447-3458
7. *Energy Environ. Sci.*, **2021**,14, 6546-6553
8. *ACS Appl. Mater. Interfaces* **2021**, 13, 44, 51917–51924
9. *Energy Environ. Sci.*, **2020**,13, 3633-3645
10. *ACS Catal.* **2021**, 11, 1, 264–270
11. *J. Colloid Interface Sci.* **2022**, 628, 306–314
12. *Angew. Chem. Int. Ed.* **2014**,53, 1378 –1381
13. *J. Am. Chem. Soc.* **2012**, 134, 22, 9054–9057
14. *ACS Appl. Energy Mater.* **2021**, 4, 2, 1053–1058

We have made the following revisions in the main text:

(1) On page 14 of the revised manuscript:

Alkaline water electrolysis employing all-in-one MEA-S outperformed most previously reported ALWE^{4, 36, 37, 38, 39, 40, 41} and AEMWE without platinum group metals (PGMs)^{14, 21, 42, 43, 44, 45, 46, 47} (**Fig. 4f**). Importantly, the performance of all-in-one MEA-S fed with 0.5 wt% KOH was comparable to that of AEMWE in 0.5 wt% KOH and pure water (**Supplementary Fig. 40**). Furthermore, our alkaline water electrolysis showed similar or higher performance compared with PEMWE reported in the literature^{48, 49, 50, 51, 52, 53} (**Table S1**).

(2) We have added this Fig. R30 as new Fig. 4f in the revised manuscript.

(3) We have added this Fig. R31 and Fig. R32 as new Supplementary Fig. 39 and Fig. 40 in the revised Supplementary Materials.

(4) We have added this Table. R5-R6 as new Supplementary Table. 3-4 in the revised Supplementary

Materials.

Comment 15:

278: Is the 30wt% KOH impedance data also available for the all in-one MEA? Why not provide both 10 and 30 wt% data sets? Not able to accurately compare impedance spectra reported after 600h stability measured in 30 wt% KOH (Fig 4e).

Response to the comments 15:

We appreciate the reviewer's valuable comments. To further highlight the merits of all-in-one MEA, and as suggested by reviewer 1, the 10 wt% data sets have been replaced by 5 wt%. Therefore, the polarization curves and impedance data of 10 and 30 wt% KOH have been provided in the revised manuscript. As shown in Fig. R33, the figure (Fig. 4e) has been improved for accurate comparison.

Fig. R33 | Nyquist impedance plots of C-MEA-CCM and all-in-one MEA-S before and after stability test.

We have made the following revisions in the main text:

- (1) On page 15 of the revised manuscript:

As shown in Fig. 5e, the R_{ohm} value of the C-MEA-CCM increased from 360 to 540 $m\Omega\ cm^2$. In contrast, the R_{ohm} value of all-on-one MEA-S operated in 5 and 10 wt% KOH showed no obvious increase.

(2) We have added this Fig.R33 as new Fig. 5e in the revised manuscript.

Comment 16:

298: The authors mention gas permeability measurements, and since 100% Faradaic efficiency is reported at both 1000 and 4000 A/cm^2 do the authors assume there is no hydrogen crossover on the anode side? This is usually only accurately determined by measuring hydrogen crossover at the anode directly.

Response to the comments 16:

We appreciate the reviewer's valuable comments. The Faradaic efficiency is measured to verify if all electron went into water electrolysis. We agree with reviewer's comments. Thus, the hydrogen crossover at the anode has been measured to evaluate gas crossover. Figure R34 shows the concentration of hydrogen in the anodic oxygen stream as a function of current density. These results, which fell between 0.07 and 0.18 vol% at 0.05–1.0 $A\ cm^{-2}$, are notably lower than reported rates of hydrogen crossover with conventional separators (Table R7).

Fig. R34 | Hydrogen crossover. The data points show the hydrogen crossover into the anodic oxygen stream

of the all-in-one MEA-S as a function of current density, at 60 °C and atmospheric pressure in 30 wt% KOH. Each data point was collected after operating the cell for 30 min at the relevant current density.

Table R7. Comparison of reported hydrogen crossover with conventional separators.

Separator	Electrolyte	Temperature / °C	Pressure / bar	Current density / A cm ⁻²	Concentration (H ₂ in O ₂) / vol.%	Reference
All-in-one MEA-S	30 wt% KOH	60	1 atm	0.05	0.18	This work
All-in-one MEA-S	30 wt% KOH	60	1 atm	0.20	0.10	This work
Zirfon PERL UTP 500	30 wt% KOH	80	10	0.40	0.21	1
PTFE/LDH	1 M KOH	80	1 atm	0.50	1.30	2
Zirfon	30 wt% KOH	80	1 atm	0.10	2.42	3
cPVAZ-30	30 wt% KOH	80	1 atm	0.10	0.32	3
cPVAZ-30	30 wt% KOH	80	1 atm	0.20	0.27	3
Zirfon	30 wt% KOH	80	0.3	0.20	1.59	4
Z80_300um	30 wt% KOH	80	0.3	0.05	1.24	4
Z80_300um	30 wt% KOH	80	0.3	0.20	0.26	4
Z80C5	10 wt% KOH	80	0.1	0.05	0.26	5
Z80C5	10 wt% KOH	80	0.1	0.20	0.21	5
Z80C5	10 wt% KOH	80	0.3	0.05	0.75	5
Nafion 117	DI water	80	1 atm	0.40	1.91	6
Nafion 212	DI water	80	1 atm	1.40	1.91	6

References in Table R7:

1. *J. Power Sources* 2014, 247, 967-974
2. *Chemical Engineering Journal* 2021, 426, 131340
3. *J. Power Sources* 2022, 524, 231059
4. *Journal of Membrane Science* 2020, 616, 118541

5. *Chemical Engineering Journal* 2022, 428, 131149

6. *J. Electrochemical Society* 2016, 163 (11) F3197-F3208

We have made the following revisions in the main text:

(1) On page 15-16 of the revised manuscript:

To assess the gas permeability of all-in-one MEAs, the anodic hydrogen contents were detected by gas chromatography. All-one MEA-S showed hydrogen contents of <0.20 vol% over the current density range of 50–1000 mA cm⁻², which was notably lower than the reported hydrogen crossover rates with conventional separators (Supplementary Fig. 42 and Table 6). Furthermore, advanced alkaline water electrolysis using all-in-one MEA-S produced H₂ with a purity of 99.981% at a high current density of 1000 mA cm⁻² (Supplementary Fig. 43). To verify if all electron were utilized in water electrolysis, the volumes of H₂ and O₂ produced by all-in-one MEA-S were measured at different current densities of 1000 and 4000 mA cm⁻² (Fig. 5f), showing almost 100% Faraday efficiency.

(2) We have added this Fig.R34 and Table R7 as new Supplementary Fig. 42 and Table 6 in the revised Supplementary Materials, respectively.

Comment 17:

336: When referring to powdery CoNiS catalyst use with the PP membrane, does the authors refer to spreading of a catalyst ink onto the membrane? The description of preparing these MEAs is missing from the method section. What is the thickness of PES porous membrane used?

Response to the comments 17:

We appreciate the reviewer's valuable comments. The MEA with powder CoNiS catalyst and PP porous membrane was prepared by conventional CCS method. The CoNiS catalyst ink was sprayed on Ni foam and carbon paper, respectively. Sorry for the confusion. The description of preparing these MEAs has been supplemented in the method section. As shown in Fig. R35, the thickness of PES porous membrane is ~20 μm.

Fig. R35 | Morphological structure characterization of PES porous membrane by SEM. (a) The surface morphology of PES porous membrane. (b-c) The cross section morphology of PES porous membrane.

We have made the following revisions in the maintext:

(1) On page 17 of legend of Fig. 5 in the revised manuscript:

Fig. 5 | Stability and increased performance of all-in-one MEAs. Morphology structure analysis after electrolysis for 600 h: (a) Cross-sectional SEM image of all-in-one MEA-S and corresponding EDS mapping. (b) Surface and (c) cross-sectional SEM images of CL. (d) Polarization curves of C-MEA-CCM and all-in-one MEA-S before and after stability tests. (e) Nyquist impedance plots of C-MEA-CCM and all-in-one MEA-S before and after stability tests. (f) Volumes of H₂ and O₂ generated by the advanced alkaline water electrolysis using the all-in-one MEA-S at different current densities of 1000 and 4000 mA cm⁻² under atmospheric pressure. (g) Bar diagram dividing over-potential into η_{ohm} , η_{kin} , and η_{mass} at different current densities of 200, 500 and 1000 mA cm⁻². (h) Polarization curves of all-in-one MEA-S and controls. C-MEA-CCS (P-CoNiS//PP//P-CoNiS) and C-MEA-CCS (P-CoNiS//PES//P-CoNiS) were prepared by loading powder CoNiS catalysts on a liquid/gas diffusion layer. C-MEA-CCS (S-CoNiS//PES//S-CoNiS) was prepared using self-supported CoNiS electrodes. (i) Voltage reductions and their origins. Waterfall plot showing the voltages reductions observed at 1 A cm⁻² (60 °C) and their sources.

(2) On page 24 of the method section in the revised manuscript:

To prepare C-MEA-CCS with self-supported electrodes, the synthesis process of the self-supported CoNiS electrode was the same as that used for all-in-one MEA-S, but replacing the porous PP membrane with Ni foam or carbon paper. The as-prepared self-supported electrodes were pressed with a membrane (such as PES, Zirfon Thin, *m*-PBI, and Sustainion X37-50) at 70 °C for 3 min at 0.1 MPa. The loadings of cathode and anode CLs using powder CoNiS catalysts on the membrane or liquid/gas diffusion layer were both 2.5 mg cm⁻².

(3) We have added this Fig.R35 as new Supplementary Fig. 45 in the revised Supplementary Materials.

Comment 18:

337: Please see Fig 4 h(i) and line 339: h (ii) notation. The reference here does not make sense or it is a typo for h & i figures?

Response to the comments 18:

Sorry for the confusion. This is a mistake. The Figure 4h(i) and Figure 4h(ii) have been corrected as Figure 5h.

We have made the following revisions in the maintext:

(1) On page 18 of the revised manuscript:

To elucidate the major factors contributing to the increased performance of all-in-one MEA-S, an alkaline electrolyzer with pristine PP porous membrane and powder CoNiS catalyst was used as a baseline (**Fig. 5h**).

Comment 19:

340: reference to Fig S40 – please double check the captions of Figures in SI. Fig S40 show images of PTFE supported PP membrane. Inconsistent with reference from main text. There is no SEM images representing PES membrane in the SI.

Response to the comments 19:

Sorry for the confusion. These mistakes have been corrected.

We have made the following revisions in the Supplementary Materials:

(1) On page 58 of in the revised Supplementary Materials:

Supplementary Fig. 45 | Morphological structure characterization of PES porous membrane by SEM.

(a) The surface morphology of PES porous membrane. (b-c) The cross section morphology of PES porous membrane.

Comment 20:

362: Supplier of the commercially used PE-reinforced PTFE membrane is not mentioned.

Response to the comments 20:

We appreciate the reviewer's valuable comments. The PE-reinforced PTFE porous membrane was obtained from Changzhou jinchun. Co., Ltd. (Changzhou, China).

We have made the following revisions in the maintext:

(2) On page 22 of the method section in the revised manuscript:

PE-reinforced PTFE porous membrane was obtained from Changzhou jinchun. Co., Ltd. (Changzhou, China).

Comment 21:

Fig 5 (b) is a comparison of different catalyst and porous membrane (PP vs PE reinforced with PTFE) investigated in this study for the purpose of finding low cost solutions. Can this be confirmed that the all-in-one MEA (CoNiS@FeNi LDH) @1 A/cm² is the PP porous membranes used in all-in-one MEA-S? The polarization curve Fig 3 (b) for all-in-one MEA-S 30wt% KOH is very similar to performance of all-in-one MEA (CoNiS@FeNi LDH) in Fig 5 (b), however the cell voltage @1 A/cm² is lower and degradation rate smaller for last mentioned catalyst variation. The MEA (CoNiS@FeNi LDH) preparation method with new catalyst is not included in the method section. Please review.

Response to the comments 21:

We appreciate the reviewer's valuable comments. Sorry for the confusion. We have corrected this mistake. The Fig. 5b has been replotted, which is exhibited in Fig. R36. In addition, the preparation method of all-in-one MEA (CoNiS@FeNi LDH) has been added in the method section.

Fig. R36 | Electrolysis performance of all-in-one MEAs. Performance was measured at 60 °C with 30 wt% KOH circulating in the anode and cathode.

We have made the following revisions in the maintext:

- (1) We have added this Fig.R36 as new Fig. 5b in the revised manuscript.
- (2) On page 23 of the method section in the revised manuscript:

Preparation of all-in-one MEA (CoNiS@FeNi LDH)

For the preparation of all-in-one MEA (CoNiS@FeNi LDH), the precursor solution was prepared by mixing Ni(NO₃)₂·6H₂O (1.6 mmol), FeSO₄·7H₂O (0.4 mmol), urea (8 mmol), and H₂O (60 mL). Then, the precursor solution containing as-prepared all-in-one MEA-S was treated by a hydrothermal method at 100 °C for 3 h.

Comment 22:

351: Please specify which all-in-one MEA the Nr 26 in Fig 5 (e) refers from those evaluated in the study.

Response to the comments 22:

We appreciate the reviewer's valuable comments. As shown in Fig. R37, the Nr 26 is all-in-one MEA (CoNiS@FeNi LDH).

Fig. R37 | The energy efficiency comparison of all-in-one MEA-S and reported literatures. Energy efficiency comparison of all-in-one MEA (CoNiS@FeNi LDH) with reported alkaline water electrolysis at 1 A cm⁻².

We have made the following revisions in the maintext:

(1) We have added this Fig.R37 as new Fig. 6 in the revised manuscript.

Comment 23:

439: Missing information on assembly: truly zero-gap tested MEA? Fig 3 (a) illustrates the configuration of the cell, but whether the MEA and LGDLs are compressed isn't clear. What is the cell size (active area of the cell) tested?

Response to the comments 23:

We appreciate the reviewer's valuable comments. The cathode carbon paper and anode Ni foam were sandwiched with the as-prepared all-in-one MEA and pressed to form a five-layered MEA at 70 °C and 0.1 MPa for 3 min. The active size of anode and cathode is 5 cm².

We have made the following revisions in the main text:

(1) On page 25 of the revised manuscript:

The electrolysis performance of MEAs was assessed in a home-made alkaline water electrolyzer with an active area of 5 cm² (**Supplementary Fig. 51**).

(2) On page 25 of the revised manuscript:

The cathode carbon paper and anode Ni foam were sandwiched with the as-prepared all-in-one MEA and pressed at 0.1 MPa and 70 °C for 3 min to form a five-layered MEA.

Comment 24:

SI: please review for spelling mistakes and control that figure captions fit the figures content.

Response to the comments 24:

We appreciate the reviewer's valuable comments. As suggested, we have carefully checked and polished our manuscript in the revised version by a native English speaker.

REVIEWERS' COMMENTS

Reviewer #1 (Remarks to the Author):

Overall, this manuscript seems to be much improved according to reviewer's comments. These results successfully described all in one MEA and its excellent performance in water electrolysis application. I would recommend this manuscript to be published in the Nature Communication.

Reviewer #2 (Remarks to the Author):

All comments have been addressed to satisfaction by authors in the revised manuscript.

The additional tomography and impedance data added in support of the interfacial resistance comparison vs pressure (CCS-MEA) as function of the electrolyte concentration (including All-in-one MEA-S) to be a valuable addition for the non-binder containing interpretation of results.

It can be added that the further figure and tables added in revised manuscript by the authors are of a high quality and fits to the standard of Nature Communications and testifies of a thoroughly conducted study. The efforts of the authors are therefor acknowledged for including additional measurements and strengthening their scientific contribution to the advanced alkaline electrolysis field. The study is certainly a valuable contribution to the field of electrolysis.

Response to reviewers' comments

Manuscript ID: NCOMMS-22-22077A

Title: Oriented intergrowth of catalyst layer in membrane electrode assembly for alkaline water electrolyzer

Authors: Lei Wan, Maobin Pang, Junfa Le, Ziang Xu, Hangyu Zhou, Qin Xu and Baoguo Wang*

We have considered reviewers' comments carefully and provided responses point by point to all questions. We also revised the original manuscript to make a new version as updated file, and the main revised part was highlighted with yellow color. The responses to the reviewers' comments are listed as the followings.

To Reviewer 1:

Referee 1:

Overall, this manuscript seems to be much improved according to reviewer's comments. These results successfully described all in one MEA and its excellent performance in water electrolysis application. I would recommend this manuscript to be published in the Nature Communication.

Response to the comments:

We thank the referee for the positive feedback.

To Reviewer 2:

Referee 2:

All comments have been addressed to satisfaction by authors in the revised manuscript.

The additional tomography and impedance data added in support of the interfacial resistance comparison vs pressure (CCS-MEA) as function of the electrolyte concentration (including All-in-one MEA-S) to be a valuable addition for the non-binder containing interpretation of results.

It can be added that the further figure and tables added in revised manuscript by the authors are of a high quality and fits to the standard of Nature Communications and testifies of a thoroughly conducted study. The efforts of the authors are therefor acknowledged for including additional measurements and strengthening their scientific contribution to the advanced alkaline electrolysis field. The study is certainly a valuable contribution to the field of electrolysis.

Response to the comments:

We thank the referee for the positive feedback.